

# Modelling freshwater quality scenarios with ecosystem-based adaptation in the headwaters of the Cantareira system, Brazil

Denise Taffarello[1], Raghavan Srinivasan [2], Guilherme Samprogna Mohor[1], João Luis B. Guimarães [3], Maria do Carmo Calijuri [1], Eduardo Mario Mendiondo[1]

[1] Sao Carlos School of Engineering, University of Sao Paulo, Sao Carlos, SP, 13566-590, Brazil
[2] Spatial Science Laboratory, Ecosystem Science and Management Department, Texas A&M University, College Station, TX 77801, USA
[3] Aquaflora Meio Ambiente, Curitiba, PR, 82100-310, Brazil

*Correspondence to*: Denise Taffarello (taffarellod@gmail.com; dt@sc.usp.br)

**Abstract.** Freshwater fluxes are influenced by the volume and quality of water at the headwaters of strategic river basins under change. Although hydrologic models provide hypothesis testing of complex dynamics occurring at river basin scales, freshwater quality modelling is still incipient at many river catchments. In Brazil, approximately only one in twenty modelling studies assesses freshwater nutrients, which limits the policies regarding hydrologic ecosystem services. This paper aims to compare freshwater quality scenarios under different land-use/land-cover (LULC) change, one of them related to the Ecosystem-based Adaptation (EbA) approach in subtropical headwaters. Using the spatially semi-distributed SWAT (Soil and Water Assessment Tool) model, nitrate and total phosphorous loads and sediments yield were modelled in Brazilian subtropical catchments ranging from 7.2 to 1037 km². Part of these catchments are eligible areas of the Brazilian PES-programmes called *Water Producer/PCJ* and *Water Conservator* in the Cantareira Water Supply System, which until the drought in 2013-15 had supplied water to 9 million people in the Sao Paulo Metropolitan Region. We considered freshwater quality modelling of three LULC scenarios, with no climate change, as: (i) recent past scenario (S1), with the historic LULC records in 1990, (ii) current land use scenario (S2), considered the LULC for the period 2010-2015 as the baseline, and (iii) future land use scenario (S2+EbA). The latter scenario proposed forest cover conversion with restoration through EbA in protected areas according to the Basin Plan of the Piracicaba-Capivari-Jundiaí (PCJ) watersheds by 2035. The three LULC scenarios were tested with the same records of rainfall and evapotranspiration observations in 2006-2014, which comprised the occurrence of extreme drought events. We propose a new index to assess hydrologic services related to the grey water footprint (greyWF) and water yield estimated. The Hydrologic Services Index (HSI), as a non-dimensional factor to compare water pollution levels (WPL) for referenced and unreferenced catchments, comprise water pollution levels for nitrate, total phosphorus and sediments. On the one hand, leaching simulations of nitrate and total phosphorous allowed for the regionalization of greyWF at different spatial scales under LULC changes. According to the critical threshold of reference catchments, HSI identified basins in less sustainable and more sustainable areas. On the other hand, conservation practices simulated through the S2+EbA scenario envisaged not only additional and viable best management practices, but also preventive decision making at the headwaters of water supply systems.





**Key words:** water quality modelling; ecosystem-based adaptation; SWAT; grey water footprint; land-use/land-
cover change; Brazil.

## 1 Introduction

Basin Plans comprise the main management tool and they plan sustainable use of water resources in both spatial
and temporal scales. For sustainable water allocation, river plans are based on accurate data on actual water
availability per basin, taking into account water needs for humans, environmental water requirements and the
basin's ability to assimilate pollution (Mekonnen et al., 2015). However, adaptive management options such as
ecosystem-based adaptation (EbA; see CBD, 2010; BFN/GIZ, 2013) and the water footprint (WF) (Hoekstra &
Chapagain, 2008) have rarely been incorporated into Brazilian Basin Plans. Moreover, integrated quali-
quantitative simulations and indicators of human appropriation of freshwater resources are seldom used in river
plans.
The WF still is a new environmental indicator in watershed plans worldwide. For example, Spain is the unique
country which uses WF as indicator in their Basin Plan (Hoekstra et al., 2017; Velázquez et al., 2011; Aldaya et
al., 2010). The clean water plan of Vancouver (June/2011) established as sustainable action the reduction of the
WF on its water resources management (MetroVancouver, 2011; Zubrycki et al., 2011). The Colombian
government was the first to publish a complete and multi sectorial evaluation of WF in its territory. Although,
this study, titled *Estudio Nacional del Agua* (Colombia, *Instituto de Hidrología, Meteorología y Estudios*
*Ambientales*, 2014), had not been included in the national water management plan, the strategic plan of
Magdalena Cauca basin incorporates the greyWF to assess agriculture pollution (Colombia, 2015, 2014 e
2010).  In Brazil, a glossary of terms released by the Brazilian National Water Agency (ANA, 2015) includes
the concept of WF to support water resources management.
The WF (Mekonnen & Hoekstra, 2015; Hoekstra et al., 2011) measures both the direct and indirect water use
within a river basin. The term water use refers to *water withdrawal*, as the consumptive use of rainwater (the
green water footprint) and of surface/groundwater (the blue water footprint), and *water pollution*, i.e., the
volume of water used to assimilate the pollutant loads (the grey water footprint (greyWF)) (see Chapagain et al.
2006; Hoekstra & Chapagain, 2008; Hoekstra et al., 2011). Given that water pollution can be considered a non-
consumptive water use (Mekonnen & Hoekstra, 2015; Hoekstra & Mekonnen, 2012), the greyWF is
advantageous by quantifying the effects of pollution by volume, instead of by concentration, in the same
measure units of consumptive uses, making water demand and availability comparable.
In addition, water footprint assessment, proposed by Hoekstra et al. (2011), comprises four phases: (1) Setting
goals, (2) Accounting, (3) Sustainability assessment, and (4) Response formulation. It is worth noting that WF
studies can be restricted to one specific activity of these phases or be related to more than one phase. At the WF
response formulation phase, the EbA options, represented by Best Management Practices (BMP) at the
catchment scale, could represent a trade-off on greyWF (Zaffani et al., 2011). That is, BMP adopted in the
catchment scale could contribute indirectly to decreasing the level of water pollution. Thus, the EbA would
compensate the greyWF of a certain river basin.



In the context of water security associated with land-use/land-cover (LULC) change, many existing conflicts
over water use could be prevented (Winemiller et al., 2016; Aldaya et al., 2010; Oki & Kanae, 2006). For
example, LULC influences water quality, which affects the supporting[1] and regulating[2] ecosystem services
(Mulder et al., 2015; MEA, 2005) and needs to be monitored for adaptive and equitable management on the river
basin scale (Taffarello et al., 2016a). In spite of discussions regarding the lack of representativeness of data used
in early studies with greyWF (Wichelns, 2015; Zhang et al., 2010; Aldaya et al., 2010; Aldaya & Llamas, 2008),
we argue that the greyWF method may account for hydrologic services and provide a multidisciplinary,
qualitative-quantitative integrated and transparent framework for better water policy decisions. Understanding
these catchment-scale ecohydrologic processes requires not only low-frequency sampling, but also automated, *in*
*situ*, high-frequency monitoring (Bieroza et al., 2014; Halliday et al., 2012), but also the use of ecohydrologic
models to protect water quality and quantity. However, freshwater quality modelling associated with EbA,
greyWF and LULC is still incipient in many river catchments. In Brazil, approximately only 5% of modelling
studies evaluate nutrients in freshwater (Bressiani et al., 2015), which limits the policies on regulating ecosystem
services.
In this research, we propose the regulating ecosystem services be addressed by the greyWF because it considers
the water volume for self-purification of receiving water bodies affected by pollutants (Zhang et al., 2010). Thus,
the hypothesis of the research is: conservation practices, addressed by BMP or EbA, and other types of land
use conversion which impact hydrology and the ecosystem services (Winemiller et al., 2016) in the catchment
and sub-basin scales. In these scales, the greyWF can evaluate the changes in the regulating hydrologic services.
Among the three water footprint components, in this study we assessed greyWF for nitrate, total phosphorous
and sediments in 20 sub-basins in the headwaters of the Cantareira Water Supply System. The aim of this study
is to compare freshwater quality scenarios, one of them related to EbA options through BMP and to assess
greyWF under different LULC changes: (S1) historic LULC of 1990; (S2) current LULC for the period 2010-
2015; and (S2+EbA) future LULC based on EbA with S2 as a baseline. This method is addressed using Nested
Catchment Experiments (NCE), (see Taffarello et al., 2016a and 2016b) at a range of scales, from small
catchments of 7.7 km$^2$ to medium-size basins of 1200 km$^2$ at subtropical headwaters responsible for the water
supply of Sao Paulo Metropolitan Region (SPMR). This paper consists of four sections. The first section
provides a brief description of the context, gap, hypothesis and our research goals. The second section describes
the simulation methods used in the watershed scale and development of three LULC scenarios. We then propose
some ecosystem-based adaptation (EbA) approaches related to water pollution. Finally, in the fourth section, we
discuss *how* the grey water footprint for nitrate or total phosphorous could be an EbA option for improving
decision-making and water security in subtropical catchments under change.

---

[1]Examples of supporting services: nutrient cycling, primary production and soil formation.
[2] Examples of regulating services: self-depuration of pollutants, climate regulation, erosion control, flood attenuation and water borne diseases.





## 2. Material and Methods

### 2.1. The case-study area

Two of the most vulnerable areas in the Brazilian South-East are the Upper Tietê (drainage area 7,390 km²) and Piracicaba-Capivari-Jundiaí - PCJ (drainage area 14,178 km²) watersheds, particularly due to their high population: 18 Mi inhabitants in Upper Tietê River basin, and 5 Mi in PCJ (Sao Paulo, 2017; IBGE, 2010). In an attempt to ensure public water supply, the government built the Cantareira System, an inter-basin transfer, in two stages: **a)** between 1968 and 1974, at the end of a 35-year period that underwent a severe drought in the Piracicaba watershed, and **b)** in 1982, with the inclusion of two additional reservoirs that regularized the increasing rainfall from the mid-1970s until 2005 (Zuffo, 2015).

The study area comprises the part of the Cantareira System that drains into the Piracicaba river and which is the headwater of the Piracicaba basin (**Figure 1**). This basin is located on the borderline of the state of Minas Gerais and Sao Paulo. This part of the water supply system, in the Piracicaba watershed, consists of three main reservoirs, named after the rivers, damming the Jaguari-Jacareí, Atibainha and Cachoeira watersheds (drainage areas are 1230 km², 392 km² and 312 km², respectively). These rivers are main tributaries of the Piracicaba river, which is a tributary of the Tiete River system, on the left bank of the Parana Basin. The Cantareira System consists of two more reservoirs out of the Piracicaba river basin, Paiva Castro and Águas Claras, which are not part of our study area. To simplify our simulations, we did not model the reservoirs´ storage nor the complex water transfer operations. The water from these five reservoirs is crucial for the water supply to South America's biggest city, Sao Paulo, as well as the Metropolitan Region of Campinas.

With respect to the water quality, the headwaters of the Cantareira System are classified as "class 1" for Jacareí, Cachoeira and Atibainha watersheds, and "class 2" for the Jaguari watershed, according to the CONAMA Resolution Nº 357/2005 (Brazil, 2005) and Sao Paulo Decree Nº 8468/1976 (Sao Paulo, 1976), which means that, with the exception of the Jaguari watershed, the others can be used with only a simple treatment. Regarding the water volume, this region has been intensely impacted by a severe and recent drought (Taffarello et al., 2016a; Escobar, 2015; Whately & Lerer, 2015; ANA, 2015; Porto & Porto, 2014). As a result of this serious water crisis, a new hydric law on the average flow of the transfer limits of the Piracicaba watershed to the Upper Tiete watershed was postponed from 2014 to May, 2017 (ANA, 2015). The Cantareira System is located in the Atlantic Forest biome, considered a conservation hotspot because of its rich biodiversity. In spite of that, 78% of the original forest cover of the Cantareira watershed has been deforested over the past 30 years (Zuffo, 2015). In 2014, the native forest cover was 10% in Extrema, 12% in Joanópolis and 21% in Nazaré Paulista (SOS Mata Atlântica/INPE, 2015). To counteract deforestation, some environmental/financial trade-offs have been developed in the Cantareira headwaters to protect downstream water quality and the regulation of water flows. These are Ecosystem-based Adaptation (EbA) initiatives, in which rural landowners receive economic incentives to conserve and/or restore riparian forests and implement soil conservation practices (see Chapter 3 of this thesis). The first Brazilian EbA approach was the *Water Conservator Project*, created in 2005 and implemented in Extrema, Minas Gerais (Richards et al., 2015; Pereira, 2013). The *Water Producer/PCJ* (Guimarães, 2013) ran from 2009 to 2014 in the Cantareira System region, which was a pioneer project in the



state of Sao Paulo that promoted: (i) forest restoration in permanent preservation areas (PPA); (ii) conservation
of remaining forest fragments; and (iii) soil conservation. As a pilot project, it focused on providing subsidies to
larger scale projects (Padovezi et al., 2013). Both projects were established through public-private partnerships,
strengthening EbA in Brazil.

**2.2. Databases and model adopted**
**Figure 2** shows the method developed and applied to assess the regulating hydrologic services through grey WF,
along with the spatial data used in this study. The simulations were enhanced by model parameterization with
qualitative and quantitative primary data (Mohor et al., 2015a; Mohor et al., 2015b; Taffarello et al. 2016b) from
six field campaigns between 2012 and 2014, in partnership with ANA, CPRM, TNC-Brazil, WWF, USP/EESC
and municipalities. This can reduce uncertainties of the model, facilitate data interpretation and provide
consistent information. We installed three data collection platforms (DCP) in catchments at Posses, Cancã and
Moinho, and level and pressure sensors in paired sub-basins *(i)* with high original vegetation cover, and *(ii)* in
basins that receive payment for ecosystem services due to participating in the *Water Producer/PCJ* project.
We obtained and organized secondary data from the region upstream of the Jaguari-Jacareí, Cachoeira and
Atibainha reservoirs. We then set up a database originating from several sources: Hidroweb (ANA, 2014); Basic
Sanitation Company of the State of Sao Paulo (SABESP); Integrated Center for Agrometeorology Information
(CIIAGRO, 2014); Department of Water and Power (DAEE); National Institute of Meteorology (INMET) from
the Center for Weather Forecasts and Climate Studies (CPTEC/INPE).
**Supplement Table S1** summarizes all hydrologic, pedological, meteorological and land-use data used as input
for the delineation and characterization of the watersheds. The topographical data used was the Digital Elevation
Model "ASTER Global DEM", 2ª version, 30-m (Tachikawa, et al., 2011), available free of charge at:
http://gdex.cr.usgs.gov/gdex/. The depressions of this DEM were fixed before making them available to users.
Worldwide uses of ecosystem service models are increasing (Posner et al., 2016). The changes in hydrologic
services can be evaluated by a wide number of models (Carvalho-Santos et al, 2016; Duku et al, 2015; Quilbé &
Rousseau, 2007), especially those more user-friendly for stakeholders and policy makers. Simulations in this
watershed-scale ecohydrologic model (Williams et al, 2008; and Borah & Bera, 2003) allow for the
quantification of important variables for ecosystem services analysis and decision-making. Some examples of
ecohydrologic models with progressive applications in Brazilian basins are SWAT (Bremer et al., 2016;
Francesconi et al., 2016; Bressiani et al., 2015), the models reviewed by de Mello et al. (2016), Integrated
Valuation of Ecosystem Services and Tradeoffs (InVEST) (Sharp, 2016; Tallis et al., 2011) and Resource
Investment Optimization System (RIOS) (Vogl et al., 2016).
Hydrologic models with freshwater quality routines (eg., QUAL-2K, QUAL-2E, SWMM, SWAT) represent the
water balance and the coupling processes of water quality. In these models, input data are converted into the
system's outputs, both quantity and quality variables, which represent the water balance and water quality
conditions. Depending on the availability of input data, the user determines whether the simulations will be



carried out over annual, monthly, daily or sub-daily time (Boithias et al., 2015) and scheduled time. As there is a
lack of water quality data on a daily basis in Brazil and considering the objectives of this study, which are
especially related to a dry period from 2013 to 2015 in the Cantareira, we chose to use the SWAT model with
monthly simulations.
The Soil and Water Assessment Tool - SWAT-TAMU (Arnold et al., 1998; Arnold and Fohrer, 2005) is a public
domain conceptual spatially semi-distributed model, widely used in ecohydrologic and/or agricultural studies at
river basin scale (Krysanova & Whyte, 2015; Krysanova & Arnold, 2008). It divides the basin into sub-basins
based on an elevation map and the sub-basins are further subdivided into *Hydrologic Response Units* (HRU).
Each HRU represents a specific combination of land use, soil type and slope class within the sub-basin. The
model includes climatic, hydrologic, soil, sediments and vegetation components, transport of nutrients,
pesticides, bacteria, pathogens, BMP and climate change in a river basin scale (Srinivasan et al., 2014;
GASSMAN et al., 2014; Arnold et al., 2012).
There have been at least 2,600 published SWAT studies (SWAT Literature Database, mid-2016). In the *SWAT*
*Purdue Conference*, held in 2015, 118 studies were presented, of which, only 8% assessed the transport of
nutrients in watersheds (SWAT Purdue, Book of Abstracts, 2015). Research using SWAT, not only for quantity
but also for water quality and ecosystem service assessments (Francesconi et al., 2016; Abbaspour et al., 2015;
Duku et al., 2015; Dagupatti & Srinivasan, 2015; Gassman et al., 2014) and also as an educational tool for
comparing hydrologic processes (Rajib et al., 2016) have increased in recent years.

**2.3. Model Set-up**

The initial model set-up used the ArcSWAT interface, integrated to ArcGIS 10.0 (Environmental Systems
Research Institute - ESRI, 2010, ArcSWAT 2012.10.15 in ArcGIS 10).
Discretization in sub-basins was carried out, where possible, at the same NCE sites of field investigations.
The delimitation of the basin using ArcSWAT requires a drainage area threshold, determined to 7.1km², dividing
the geographical space to represent the 17 sampling sites in the research field as sub-basins, plus the limits of the
three reservoirs´ drainage areas, which resulted in 20 sub-basins (**Table 1 and Figure 1b**). We highlight that the
basin was designed up to the confluence of the Jaguari and Atibaia Rivers, forming the Piracicaba river, to
integrate all areas of interest in the same SWAT project.
The definition of the HRU was carried out using soil maps of the state of São Paulo. (Oliveira, 1999) and land
use maps were developed by Molin (2014; et al. 2015) from LANDSAT 5 TM imagery for 2010, using a
1:60,000 scale. The procedure defined 49 HRUs inside the 20 sub-basins, i.e. 49 different combinations of soil
type, soil cover and slope classes in our study area.
Next, we adapted the land use map developed by Guimarães (2013), which represents a 2010 land use scenario
for the Cantareira System restoring the most fragile degraded parcels (greatest potential for sediment
production), to agree with the land use classes of Molin (2014). Additionally, we assumed that the Second
Scenario of Guimarães (2013), who used the INVEST model to provide the ecological restoration benefits in the




214 Cantareira System, could be achieved in 2035, considering the investments provided in the PCJ River Plan

215 (Cobrape, 2011) to recover riparian forests in the Cantareira System. It is worth mentioning that in the PCJ Basin

216 Plan, this is called "Trend Scenario". As in the region the restoration of riparian forests is mostly due to Water-

217 PES projects, which was recognized as an Ecosystem-based Adaptation (EbA) (CBD, 2010; BFN/GIZ, 2013;

218 Taffarello et al., submitted), we identify the third scenario as S2+EbA. Thus, **Figure 3** shows the land-use

219 changes over time.

220 In the "Trend Scenario" (PCJ-COBRAPE, 2011), the municipalities covered by the Cantareira System could

221 reach a 98% collection rate, collected sewage treatment rate of 100% and $BOD_{5,20}$ removal efficiency of 95%

222 (PCJ-COBRAPE, 2011). We emphasize that in Brazil the current allowed discharge is only based on the $BOD_{5,20}$

223 parameter. Some studies have suggested including other parameters such as dissolved oxygen, nitrate and

224 phosphate polluting loads, as well as sediments to assess the water quality (Cruz, 2015; Cunha et al., 2014).

225 Regarding the treatment costs for drinking water supply, ecosystem-based adaptation options, such as watershed

226 restoration, seem to be more cost-effective than many technologies for water treatment (Cunha; Sabogal-Paz &

227 Dodds, 2016).

228

### 2.4. Calibration & validation

230 We used the SWAT CUP 5.1.6.2 interfaces and Sequential Uncertainty Fitting (SUFI-2) algorithm for

231 calibrating the quantity and quality parameters and also for validating the simulations in the sub-basins.

232 Quantitative calibration was performed in stations that had more than two full years of observed data, i.e., 8

233 stations, namely: Posses outlet, F23, F24, F25B, F28, Atibainha reservoir, Cachoeira reservoir, Jaguari and

234 Jacarei reservoirs (**Table 2**). A common test period for all LULC scenarios was selected, in our case, the test

235 period ranges from 01 Jan, 2006 to 30 June, 2014. This period has the rain-anomaly of drought conditions from

236 2013 to 2014.

237 The calibration period was from October, 2007 to September, 2009, the only period with observed data in all of

238 the above 8 stations. Validation took place from January, 2006 to September, 2007 and from October, 2009 to

239 June, 2014. Calibration and validation of SWAT at the stations with over 2 years of data were rated as "good",

240 according to the classification by Moriasi et al. (2007), since the Nash-Sutcliffe Efficiency (NSE) criterion (Nash

241 & Sutcliffe, 1970) was greater than 0.65, except for the Posses outlet, which presented the logarithmic Nash-

242 Sutcliffe (NSElog) (using the logarithm of streamflow, a criterion that gives greater weight to smaller flow rates)

243 of less than 0.5, rated as "unsatisfactory".The Percent Bias (Pbias) statistics indicates the bias percentage of

244 simulated flows relative to the observed flows (Gupta et al., 1999). Thus, when the Pbias value is closer to zero,

245 it results in a better representation of the basin, and in lower estimate tendencies (Moriasi et al., 2007). As a

246 general rule, if | Pbias | < 10%, it means a very good fit; 10% < | Pbias |< 15%, good; 15% < | Pbias | < 25%,

247 satisfactory and | Pbias | > 25%, the model is inappropriate. On the other hand, the NSE coefficient translates the

248 application efficiency of the model into more accurate predictions of flood flows, using the classification: NSE >



0.65 the model is rated as very good; 0.54 < NSE < 0.65 the model is rated as good and between 0.5 and 0.54, it
is rated as satisfactory.
In the results obtained for different basin scales (**Figure 4**), the Pbias and NSE coefficients (including NSE of
logarithms) indicate adequate quantitative adjustments. As the SWAT simulations include more than 200
parameters, based on research from the literature (Duku et al., 2015; Bressiani et al., 2015; Arnold et al., 2012;
Garbossa et al., 2011), we selected approximately 10 parameters (see **Table 3**) to complete the calibration to
simulate streamflow processes and nutrient dynamics. These parameters refer to key processes which represent
soil water storage, infiltration, evapotranspiration, flow channel, boundary conditions (see Mohor et al., 2015b)
and main water quality processes at hillslopes. Although our calibration is mainly focused on water yield as total
runoff, freshwater quality features through pollutant loads were performed in the scenarios.
Moreover, to reduce the uncertainty of our predictions, we used approximately 2500 primary data derived from
an earlier stage of this research (Taffarello et al., 2016a). Our decision to complement field and laboratory
methods with computational tools in order to understand the behaviour of basins is justified by Tucci (1998),
who explains the need for flow and other hydrologic variables measurements, in addition to using the models,
because *"no methodology can increase the existing information in the data, but can better extract the existing*
*information."* As a parametrization result of field investigations and ecohydrologic modelling, **Figure 5** shows
parts of the calibrated model performance (lines) against field observations (dots with experimental uncertainty)
for flow discharges, nitrate and total phosphorus loads for catchment areas ranging from 7.1 to 508 km². Finally,
other water quality variables were studied based on data from field sampling.
We highlight some SWAT model limitations when we compare the simulated to observed water flows,
especially in the dry season. For example, when the model was discretized on a daily resolution, the adherence
level between the observed and simulated flows was considered good. However, the model did not fit well to
observed values during the drought period (Feb/2014-May/2014). These differences were more significant for
water quality parameters, such as nitrate and total phosphorous. We point out that the macronutrient loads found
in May, 2014 were clearly higher than the loads we found in previous sampling, which occurred in wetter
periods (Taffarello et al. 2016). For the sample collected in May, the model significantly underestimated the
pollutant loads of nitrate. This behaviour, arising from the recent and most severe drought faced by the
Cantareira System (Nobre et al., 2016; Marengo et al., 2016; Taffarello et al. 2016; Escobar, 2015; The
Economist, 2015; Porto & Porto, 2014), shows a need for improving the SWAT model performance if one has
extreme events as the main goal, especially to capture nonlinearities having impacts on regulating ecosystem
services.

**2.5. The scenarios and a new index for hydrologic service assessment**

Differences in flow rates and water quality (for the variables nitrate, phosphate, $BOD_{5,20}$, turbidity and faecal
coliforms) for the 20 sub-basins were evaluated using flow and load duration curves for the three scenarios
proposed in this study: (i) *recent past scenario* (S1), including the recorded past events for land use in 1990, (ii)
*current land use scenario* (S2), which considered land uses for the 2010-2015 period as the baseline, and (iii)
*future land use scenario* (S2+EbA), supposing a forest cover conversion in the protected areas, through EbA





options, according to the PCJ River Basin Plan by 2035. Using these curves, from the methodology shown by
Hoekstra et al. (2011), and based on Duku et al. (2015) and Cunha et al. (2012), we estimated the grey water
footprint (greyWF). Next, we developed a new ecohydrologic index to assess the regulating hydrologic services
in relation to the greyWF.
This new indicator encompasses the former theory related to environmental sustainability of the greyWF,
according to Hoekstra et al. (2011). In this study, as a relevant local impact indicator, Hoekstra et al. (2011)
proposed to calculate the 'water pollution level' (WPL) within the catchment, which measures the degree of
pollution. WPL is defined as a fraction of the waste assimilation capacity consumed and calculated by taking the
ratio of the total of greyWF in a catchment ($\sum WF_{grey}$) to the actual runoff from that catchment ($R_{act}$), or, in a
proxy manner, the water yield or mean water yield or long-term period ($Q_{lp}$). This assumption is that a water
pollution level of 100 per cent means that the waste assimilation capacity has been fully consumed. Furthermore,
this approach assumes that when WPL exceeds 100 %, environmental standards are violated, such as:
$$WPL\,[x,t] = \frac{\sum WF_{grey}[x,t]}{R_{act}[x,t]},$$

299         (1)

It is worth mentioning that for some experts, the aforementioned equation can overestimate the flow necessary
to dilute pollutants. For that reason, new insights of composite indicators or thresholds are recommended, as
follows.
The above assumption could overestimate WPL because it would fail considering the combined capacity of
water to assimilate multiple pollutants (Hoekstra et al., 2012; Smakhtin et al., 2005). Conversely, in this study,
we define an alternative indicator related to the three following fundamentals. First, the WPL should be extended
to a composite index, thereby representing weights of each pollutant related to the actual runoff, here as a proxy
of long-term runoff, i.e.:

$$WPL_{composite}[x,t] = \frac{\sum\{w[x,t]*WF_{grey}[x,t]\}}{R_{act}[x,t] \cong Q_{lp}[x,t]},$$

310         (2)

$$\sum w[x,t] = 1$$
$$0 \le w[x,t] \le 1$$

For this new equation, weights should be assessed, either from field experiments or even from simulation
outputs. Second, we define a threshold value of WPL composite regarding the reference catchments in non-
developed conditions which suggest more conservation conditions among other catchments of the same region,
as *WPL_{reference}*. For this study, we selected *Domithildes* catchment as the reference catchment with conservancy
measures. From this reference catchment, we define the composite reference index for the water pollution level
as *WPL_{composite,ref}* and, derived from it, the Hydrologic Service Index, as a non-dimensional factor of comparison
between WPL for reference and non-reference catchments, as follows:



$$HSI[x,t]_{greyWF} = \frac{WPL\,[x,t] - WPL_{composite,ref}}{WPL_{composite,ref}},$$

322       (3)

### 3. Results and Discussion

In the following section, we present the results from field observations, useful not only for ecohydrologic
parameterization, but also to elucidate features regarding greyWF and hydrologic services. Next, we compare the
water yield and greyWF outputs from simulations under LULC scenarios, including EbA options, to finally
propose a new hydrologic services indicator.

### 3.1. Data from field sampling

Some of the water quality and quantity variables from our freshwater monitoring are useful to assess the
hydrologic services, thus they are presented in **Table 4**. These variables were selected due to their relationship
with anthropic impacts on the water bodies and because of their importance for sanitation
Among the water quality variables sampled in the field step of the research (see Taffarello et al., 2016a;
Taffarello et al., 2016b), we highlight turbidity because it indicates a proxy estimation about the total suspended
solids in lotic environments (UNEP, 2008), related to the LULC conversion and reflects the changes in the
hydrologic services. **Figure 6** shows the direct correlation between turbidity and size of the sub-basins. Turbidity
can indirectly indicate anthropic impacts in streams and rivers (Martinelli et al., 1999). The lower turbidity mean
values were observed in two more conserved sub-basins (which presented higher amounts of forest remnants): 2
NTU in the *reference Cancã catchment* (Domithildes) and 5 NTU in *Upper Posses*. Other conserved subbasins
also presented low mean values of turbidity (< 6.5 NTU): *intervention Cancã* catchment (5 NTU), and
*Cachoeira dos Pretos* (6 NTU). We found the highest turbidity, above 40 NTU which is considered the
maximum established water quality standard for Brazilian Class 1 (BRASIL, 2005): at *Parque de Eventos* (283
NTU), at *F23* (180 NTU) and at *Salto outlet* (160 NTU). However, these three sampling sites are located at
water bodies of Class 2, where the maximum turbidity allowed is up to 100 NTU (BRAZIL, 2005). Due to these
areas have the highest urbanization among the sampled sites, they are in non-compliance with Brazilian
environmental standards. Arroio Júnior (2013) found a decreasing relation between turbidity and drainage areas
in another catchment located in Sao Paulo state.
Temporal turbidity patterns show that on the one hand in 11 out of 17 monitored sites, the higher values of
turbidity occurred in December, 2013, the only field campaign with significant precipitation (35.3 mm) and with
a higher antecedent precipitation index (API = 123.7mm). This can be due to carrying allochthone particles,
which are drained into rivers by precipitation. Similarly, Arroio Júnior (2013) also observed higher turbidity in
the rainy season (December, 2012) which can lead to erosive processes. On the other hand, Zaffani et al. (2015)
showed that turbidity did not vary over the hydrologic year in medium-size, rural and peri-urban watersheds
ranging from 1 to 242 km$^2$. In this case, other factors may have had an influence, such as deforestation, seasonal
variability, soil use type, sewage and mining (CETESB, 2015; Tundisi, 2014).





Otherwise, we found a positive relationship between nitrate concentrations and both discharge and mean water
level (**Figure 7**). It can be inferred that higher concentrations of macronutrients would be found in downstream
areas. This trend can be associated to the nutrient migration (Cunha et al., 2013) and land-use change (Zaffani et
al., 2015), as well as point source pollution. In addition, the absence of the riparian forest in 70% of protected
area (36.844 ha) of the Cantareira System (Guimarães, 2013) can increase the sediment transport from riparian
areas to rivers and make pollutant filtration more difficult, leading to higher nitrate concentrations downstream.

**3.2. LULC change scenarios**
The variations in LULC affect freshwater quality which, in turn, affect the dynamics of aquatic ecosystems
(Zaffani et al., 2015; Botelho et al., 2013; Hamel et al., 2013; Bach & Ostrowski, 2013; Kaiser et al., 2013).
These changes impact the hydrologic services, especially regulating and supporting ecosystem services (Mulder
et al., 2015; Molin et al., 2017).
The LULC of each sub-basin, according to a past-condition scenario (S1, in 1990), a present-condition (S2, in
2010) and a future (S2+Eba, in 2035) LULC scenario, using the same weather input datafiles, is shown in **Table**
**5**.
The sub-basins that contain the Jaguari and Jacareí reservoirs, which are connected to a channel, have a
significant percentage of surface waters, occupying 1% of sub-basin 10 and 20% of sub-basin 15. We evaluated
the effects of LULC change scenarios in 20 catchments in the Jaguari, Cachoeira and Moinho sub-basins, South-
East Brazil. Concerning the land-use change, the main soil use 25 years ago was: pasture (in 50% of the sub-
basins) and native vegetation (in 45% of the sub-basins). According to ISA (2012) and Molin (2014), the 5% of
the remaining area were divided into vegetables, eucalyptus, sparse human settlements, bare soil and mining.
The main activity in the past (1990) was extensive cattle raising for milk production by small producers in the
region (ANA, 2012; Veiga Neto, 2008).
In the S2 Scenario (2010), the main soil use is pasture in 58% of the sub-basins and forest in 40% of them. From
1990 to 2010, there was a significant conversion of soil cover, with a slow reduction of pasture areas (-2%) and
native remnants (-5%) and with a progressive increase of eucalyptus (*Eucalyptus* sp.), an exotic forest in Brazil.
Eucalypt soil use varied from +1%, within *Posses* up to +31% in the *Chalé Ponto Verde* sub-basin in 2010.
Eucalyptus cover, however, did not achieve 10% of the soil uses in any of the simulated sub-basins in 1990. In
the third scenario (S2 + EbA), we hypothesized incentives of public policies for forest conservation and
restoration, due to the strengthening of EbA in the Cantareira System. This could lead to an increase in native
vegetation reaching percentages of 15% in the *Posses outlet* and 69% in the *F28 sub-basin*. In this scenario, the
higher percentages of native vegetation would occur in the sub-basins *F28, Upper Jaguari and Cachoeira dos*
*Pretos*.
By assessing the temporal trends of increment or reduction of native remnants, we examined the periods 1990-
2010 versus 2010-2035. From 1990 to 2010, the percentage of forest increased by 50% in the *Domithildes* sub-
basin, which was the reference catchment of the Water Producer/PCJ project, (see Taffarello et al., 2016a),




*Moinho, Cachoeira dos Pretos, F34, B. Jacareí, B. Atibainha, B. Cachoeira, Pq Eventos, F25B* and *B. Jaguari*
(**Figure 9**). Concerning the period from 2010-2035, the model was set up considering an increase in native
vegetation in all sub-basins from forest remnants in 2010, and from the new BMP practices of reforestation with
native species in 20 sub-basins by 2035 (**Figure 9**). The hydro-services in the *Posses* and *Salto* catchments and
in the *Cachoeira* sub-basin will be increased by 2035 as a function of the efforts on EbA which currently exist in
the region (Richards et al., 2017; Richards et al., 2015; Santos, 2014).
Despite this general increase in native forest cover, we highlight the deforestation which occurred in the *F23*
sub-basin in the Camanducaia river. Currently, although the basin has 34% of native forest cover, this rate has
tended to decrease since 1990. The *F23 outlet* (sub-basin 2) had 37% of native forest cover in 1990, which then
became 34 % in 2010 and the S2+EbA Scenario predicts that F23 could reach 36.2% of native forest by 2035,
returning to the percentages found in 1990. Another critical situation is the *Posses outlet* (SWAT sub-basin 6):
despite the conservation efforts which have been made in the region through the *Water Conservation* project (see
Richards et al., 2015; Santos, 2014; Pereira, 2013), the current percentage of native remnants is 13%, which can
become 16% in 2035, however not achieving the rate in 1990 (22%). This can potentially disrupt the regulating
and provision hydrologic services provided by Posses sub-basin and needs to be evaluated in depth.
Next, spatio-temporal patterns of the main soil uses which compete with forest cover are analysed: pasture and
eucalyptus. First, related to pasture, it can be observed that it was the main use in the past in 60% of the sub-
basins (in 1990) and, currently, it has become the majority LULC, approximately 40%. Our scenarios indicate
that due to EbA strengthening, encouraging the links between environmental conservation and forest restoration,
20% of the sub-basins could be mainly occupied by pasture (sub-basins 2, 4, 6 and 7). This rate is reasonable,
considering rural sub-basins. Moreover, the reduction in pasture in the Cantareira System was more evident in
the 1990-2010 period than in the 2010-2035 scenario. This can be explained by, at least, three factors: i) rural
landowners awareness of the relevance of converting pasture to native forest to generate and maintain ecosystem
services in the Cantareira System (Saad, 2016; Extrema, 2015; Mota da Silva, 2014; Padovezi et al., 2013;
Gonçalvez, 2013; Veiga-Neto, 2008); ii) seasonal changes in the ecosystem structure which can increase the
ecosystem resilience (Mulder et al., 2015) and an observed significant increase, mainly in the 1990-2010 period,
of non-native species plantations.
Second, regarding the eucalyptus cover, the future scenario shows an increasing threat to the regulating and
supporting services as a result of the exotic forest in expansion.  In 2035, eucalyptus cover may include, on
average, 12% of the total area of the 20 catchments studied here. This is significant in comparison with 10% in
2010 and only 2% in 1990 for the same catchments. The scenario for 2035 shows that the maintenance of
hydrologic services deserves attention, because eucalyptus monoculture can potentially impact not only the
headwaters, but entire landscapes, threatening the ecosystem dynamics. Moreover, these plantations, with an
average wood yield of 50 to 60 m$^3$ of *Urograndis* per hectare, need high quantities of agrochemicals, due to the
low diversity of the population and low adaptation to climate change (Kageyama & dos Santos, 2015). In short,
here we highlight the threat on biodiversity that has been brought by alien species in headwaters and the changes
that it can promote on native species (Hulme & Le Roux, 2016) which, in turn, impact the ecosystem services.





Considering the river basin as the management unit, the soil uses affect not only the quantity, but also the quality
of water resources. Thus, we analyse water and nutrient yields, intra-annual regime and duration curves, both in
quantity and quality of the pollutants, in the following topics.
**3.3. Water yield as a function of soil cover**
In hydrologic methodologies, the use of expressive variable numbers in describing the hydrologic regime for
riparian ecosystems conservation is valuable (Collischonn et al., 2005). In this context, simulations are assessed
by analysing the balance of hydrologic cycle components at determined spatial and temporal scales. The results
were analysed, on the one hand, considering regional comparisons of the size of the drainage areas and, on the
other hand, the hydrologic function that characterizes the water and nutrient availability.
The selection of the hydrologic function that indicates the water availability may be related to the
representativeness of the environmental and physical processes that occur in the catchment scale dynamically
(Cruz & Tucci, 2008). In this research, we chose to use quali-quantitative duration curves for integrated
assessment of availability and quality of water. The flow-and-load duration curve, comparable to histograms of
relative cumulative frequencies of flows and loads of a waterbody, is a simple and important analysis in
hydrology (Collischonn & Dornelles, 2013). In quantitative terms, the flow duration curve shows the
probabilistic temporal distribution of water availability (Cruz & Silveira, 2007), relating the flow in the river
cross section to the percentage of time in which it is equalled or exceeded (Cruz & Tucci, 2008).
The three scenarios S1, S2 and S2+EbA resulted in different flow values for the 20 sub-basins (**Figure 10**).
Based on the arithmetic mean of time series of monthly water yields, related to catchment areas, and assessed for
all modelled sub-basins (N=20), the results show average values of water yield: $31.4 \pm 25.2$ L/s/km² for S1
(1990), $14.9 \pm 11.5$ L/s/km² for S2 (2010) and $21.4 \pm 15.3$ L/s/km² for S2+EbA (2035), respectively. This very
high variation can be due to the complexity of river basin systems and the various sources of uncertainty in the
representation of ecohydrologic processes.
The three scenarios analysed and the ecohydrologic monitoring provide different types of information for the
same catchments. But how can we integrate the relative importance of information from each source (Kapustka
& Landis, 2010)? A detailed study showing the relationship between sensitivity (and uncertainty) of analysis and
the effectiveness of Water-PES should be carried out.
For a while, the decrease of -52.4% in water yield between S1 (1990) and S2 (2010) scenarios (= (14.9-
31.3)/31.3·100) could be due to marginal increases of eucalyptus cover. In fact, from 1990 to 2010, eucalyptus
cover increased +6.8 % in total land cover, but +181% in relative terms. Another possible explanation is the
decrease in native vegetation from 1990 to 2010, with -1.8 % in total land cover, but -4.3%, in relative terms.
In parallel, we evaluated the water yield. Thus, the flow-and-load duration curves summarize the flow and
pollutant load variability, thereby showing potential links and impacts for aquatic ecosystem sustainability
(Cunha et al., 2012; Cruz & Tucci, 2008). From these curves, we obtained two different behaviours for the
studied sub-basins (**Figure 10**):





**Behaviour I:** the water yield in 2010 reduced in relation to 1990 and the water yield in 2035 might exceed the 1990 levels. The examples are: *Upper Jaguari*, *Cachoeira* sub-basin (including the *Cachoeira dos Pretos, Chalé Ponto Verde, Ponte Cachoeira, F24 outlet)* and *Moinho* catchments;

**Behaviour II:** the water yield after 2010 was reduced until 2035 and this water yield recuperation was not possible for the values in 1990. Examples, in decreasing size of drainage areas, are: *Atibainha, B. Jaguari, F25B, Parque de Eventos, F23, B.Atibainha, F34, F30, Salto, Posses Outlet, Domithildes, Portal das Estrelas (Middle Posses)*.

On the one hand, according to **Figure 11**, the water yield of S1 is inversely proportional to the land use of mixed forest cover. The water yield in S2 indicates a constant value of approximately 17 L/s/km$^2$. Moreover, for the S2+EbA scenario, which incorporates the EbA approach through BMP, the water yield is approximately 17 L/s/km$^2$, but with a slight increase in the water yield when the percentage of forest cover is higher than 50%. Presumably, this slight increase in the water yield would be related to the type of best management practices (BMP) of the recovery forests, which still did not achieve evapotranspiration rates of the climax stage. In the riparian forest recovery, evapotranspiration rates are lower and, thus, a greater amount of precipitation reaches the soil and rivers through the canopy. This process could benefit other hydrologic components, such as runoff, increasing water flows into the rivers. This effect can possibly explain the **behaviour I** catchments (see **Fig. 10**).

On the other hand, we observed in *Posses, Salto, Jaguari, Cancã* and *Atibainha* catchments an inverse situation (**behaviour II**). This effect can be related to the hydrologic response produced by: (a) type of catchment; (b) size of catchment; (c) the low soil moisture in the red-yellow latosol (Embrapa, 2016), which did not favour high evapotranspiration rates; (d) the riparian forest, originating from the EbA or Water-PES actions, that should still be at the initial stages, not achieving a climax in 20 years (this explanation therefore assumes that the baseline of PES actions was in 2015, although there are examples of restored forests in Extrema-MG with high evapotranspiration rates, as can usually be found in climax forests); and (e) unpredictability, non-linearity and uncertainty (Ferraz et al., 2013; Lima & Zakia, 2006).

The role of the forest in the hydrologic cycle in river basin scales has been debated for centuries. Riparian native forests, eucalyptus and riparian forests in recuperation (shown here as orchard) have different hydrologic responses. There is still a lack of knowledge regarding the influence of different types and phases of vegetation on the hydrologic processes. Bayer (2014) found that the vegetation height and leaf area index are inversely proportional to the water flows, which corroborate previous studies (Hibbert, 1967). Riparian forest restoration increases the mean evapotranspiration, reducing the water yield (Molin, 2014; Salemi et al., 2012; Lima & Zakia, 2006; Andreassian, 2004). Restoration increases the water storage capability into the catchment throughout the riparian zone, contributing to the higher water flow in the dry season (Lima & Zakia, 2000). This can lead to unexpected results regarding water yield. Furthermore, at small catchments of temperate climate, researchers estimated that deforestation in 40% of the catchments would increase the runoff of $130 \pm 89$ mm.year$^{-1}$ considering the entire water cycle in the catchment scale (Collischonn & Dornelles, 2013). In addition, there is high dispersion in the results based monitoring (usually, in paired catchments or Nested Catchment Experiment - NCE), which makes it more difficult to predict the flow as a result of soil use





conversion. Similarly, we found high dispersion in the comparison between water yields *versus* different land
cover in 20 sub-basins of the subtropical climate (**Figure 11**).
BMP have been in progress since 2005 in the *Posses Outlet* (sub-basin 6, **Table 5**) and *Middle Posses (Portal*
*das Estrelas*, Nº 7), and since 2009 in *Domithildes, F30* and *Moinho* catchments (Subbasins 9, 11 and 20,
respectively). These BMP originated from the *Water Conservator* and *Water Producer/PCJ* projects. In these
cases, we recommend that public agencies take care when defending PES as inductors of more water availability
(ANA, 2013). Parts of these results and previous investigations, which were made through NCE (Taffarello et
al., 2016a), point out the opposite, i.e., in the more conserved catchments, we found lower water yields. Despite
the fact that there are many Water-PES programs in Brazil (Pagiola, von Glehn & Taffarello, 2013; Guedes &
Seehusen, 2011), measurements of the effect on water yield under forest restoration are still lacking in tropical
and subtropical conditions (Taffarello et al., 2016a; Salemi et al., 2012). However, the benefits of riparian forests
on water quality, margin stability, reduction of water erosion and silting are clear in the scientific literature
(Santos, 2014; dos Santos et al., 2014; Studinski et al., 2012; Udawatta et al., 2010).

**3.4. Relationships between land-use/land-cover change and grey water footprint**
For an integrated assessment of hydro-services, we analysed the spatio-temporal conditions of load production at
the sub-basin scale. As we studied rural sub-basins, water pollution is mainly produced by diffuse sources, such
as fertilizers and agrochemicals. In this context, we evaluated the evolution of greyWF to show nitrate ($N-NO_3$),
total phosphorus (TP) and sediment (Sed) yields (indicated by turbidity) of scenarios S1, S2 and S2+EbA. First,
we calculated the nitrate loads generated from the 20 sub-basins in the three scenarios. Second, we did the same
for total phosphorous loads and sediment yields. Third, considering the river regime, we calculated the greyWF
for nitrate, total phosphorous and sediments in each sub-basin to develop a new composite index that assesses
the sustainability of hydrologic services.
Concerning nitrate, the sampled concentrations were low. In addition, SWAT simulations also brought very low
outputs, and the greyWF-$NO_3$ varied from 0.11 $L/s/km^2$ (in *Atibainha* subbasin in S2 (2010) scenario) to 2.83
$L/s/km^2$ (in *Middle Posses* catchment, *Portal das Estrelas*, under S2+EbA (2035) scenario). Considering
Brazilian water quality standards for nitrate, the maximum allowed concentration is 10 mg/L (Brasil, 2005).
These low amounts of nitrate loads make the greyWF-$NO_3$ fall to low values in the three scenarios analysed
(between 1 and 10%; **Figure 12a**).
In relation to total phosphorous (TP), the load duration curves from S1, S2 and S2+EbA scenarios showed
disparities. For example, the greyWF-TP decreased in all sub-basins between 1990, 2010 and 2035. From 2010
to 2035, the model predicts a new behaviour for the greyWF-TP.
Results of the greyWF for TP, $NO_3$ and sediments enabled us to infer some regionalization for nutrient loads.
Among the 20 sub-basins studied, we selected 2 sub-basins as study cases to illustrate the links between LULC
and greyWF: (1) the *Upper Jaguari* and (2) *Domithildes*.





### 3.4.1 Case study I: Upper Jaguari sub-basin

The Upper Jaguari has 302 km² and is the second most upstream sub-basin within the Cantareira System (downstream of only *F28* sub-basin, with 277 km²). Comparing scenario 1990 (S1) and 2010 (S2), the results showed evidence that the native forest decayed approx. 10 %. Indeed, scenario 2035 (S2+EbA) still assumes a very small decrease in the native forest. This decrease may be due to the increase in secondary forests by BMP, which could stabilise the native forest LULC by 70% until 2035. The mean annual simulated water yields, in spite of high variability of simulated scenarios, pointed out values of 18 L.s⁻¹.km² (1990, S1), 13 L/s/km² (2010, S2) and 21 L/s/km² (for 2035, S2+EbA). Variabilities are related to hydrologic conditions simulated in the test period from 2006 to 2014. In turn, this test period was selected due to high availability of rainfall stations under operation, which would potentially better perform distributed modelling at several sub-basins using SWAT. In summary, for the three scenarios simulated, the relationships between the native forest cover and mean water yield are different from each other. For scenario S1 (1990), the higher the native forest cover, the lower the water yield. This scenario behaviour is extended at experimental sites, and even extensively documented in the literature (Salemi et al, 2012; Smarthust et al., 2012, Collischon & Dornelles, 2013). In turn, for scenario S2 (2010) the water yield seems not fully related to native forest LULC, oscillating around an average value of 18 L/s/km². In scenario S2+EbA (2035), however, there is a slight increase in water yield when native forest cover is higher than 50%. This proportional relation between water yield and forest cover in the S2+EbA is both controversial and contrary to results published by some authors (e.g. Collischonn & Dornelles, 2013; Salemi et al., 2012). For example, monitoring data shows a reduction in the water yield with higher native forest land cover (Taffarello et al., 2016a). Salemi and co-authors, in a review on the effect of riparian forest on water yield, found that riparian vegetation cover decreases water yield on a daily to annual basis.

Furthermore, the greyWF-NO$_3$ of the *Upper Jaguari* basin showed 0.14 L/s/km² for scenario S1 (1990), increased to 0.23 L/s/km² for scenario S2 (2010) and could grow to ca. 0.54 L/s/km² in S2+EbA scenario (in 2035). However, this result is different from the one expected in the hypothesis testing through modelling. The null hypothesis states that increasing native forest cover is correlated to decreasing nutrient loads flowing to streams. The results, modelled by SWAT, predicted an increase in the greyWF by 2035. The simulated increase in the native forest (approx. +5%) appears to be insufficient for buffering nitrogen loads from animal excrements such as mammals or zooplankton. For a more in-depth analysis, other factors that influence the greyWF should be evaluated thoroughly.

Concerning the greyWF in the *Upper Jaguari* sub-basin in the S2+EbA (2035) scenario, SWAT outputs assessed ca. 0.1L/s/km² related to total phosphorous (greyWF-TP) and 0 L/s/km² for sediments (greyWF-Sed). In this sub-basin, diffuse pollution from nitrates would be 5 times higher than pollution from TP. Adaptive management is needed to avoid future problems of eutrophication caused by excessive nitrogen in waters. As nitrogen is highly mobile in freshwater and terrestrial ecosystems, surface water nitrate isotopes could be used to monitor nitrogen variations in catchment-scale attenuation, as proposed by Wells et al. (2016). In this context, the calculus of greyWF for nitrate, using nitrate isotopes ($\delta^{15}$N and $\delta^{18}$O of NO$_3^{-2}$), could be a useful tool to understand spatial and temporal variations in nitrogen export throughout the catchments.



### 3.4.2 Case study II: Domithildes headwater

The *Domithildes* catchment (9.9 km$^2$) is located in the *Cancã* catchment. Similar to *Upper Jaguari, Domithildes* is one of the most conserved sub-basins, mainly with native forests. The native forest fraction remained constant (see **Figure 14**) from S1 (51% in 1990) to S2 (52% in 2010). However, unlike the *Upper Jaguari* sub-basin (see **Figure 13**), native vegetation could increase by 56% in S2+EbA (2035). Due to the fact that Domithildes was adopted as a reference basin for Water Producer/PCJ, the augmented fraction of native forest by 2035 could show an increase of secondary forest.

Regarding water yield, the *Domithildes* catchment was classified as a second type of 'subbasin behaviour' (Section 3.3). There is a positive increment of water yield between 2010 (~18 L/s/km$^2$) and 2035 (~23 L/s/km$^2$), although this situation may not achieve values obtained for S1 conditions in 1990 (~ 29 L/s/km$^2$).

Other factors, such as native vegetation, could influence the hydrologic cycle at the *Domithildes* catchment, decreasing water yields in the 2010 scenario (S2). One explanation of this water yield decrease could be the positive LULC of *Eucalyptus sp.* to +5% in 2010 (S2). Regardless of other factors, +1% of eucalyptus land-use fraction in *Domithildes* will represent -2 L/s/km$^2$ of water yield, or -63 mm per year, in the same range of results reported by Salemi (2012) and close to Semthurst et al (2015).

Comparing seasonal water yields, the results showed higher variability around monthly flow averages for the S2+EbA (2035) scenario. These deviations in monthly flows of the 2035 scenario were higher in wetter months between November and March. The regulation of water yield, in both rainy and dry conditions, is more effective when quantified through variance (Molin, 2014). In spite of these uncertainties, scenarios modelled by SWAT estimated the highest mean monthly water yield in February (38 L/s/km$^2$) and the lowest mean monthly water yield in September and October (8 L/s/km$^2$). On the one hand, the results showed that a growing rate of native vegetation LULC since 2010 would serve to attenuate both e-flows peaks, especially in the rainy season (see flow duration curves), and pollutant filtration (see duration curves of N-NO$_3$ loads). On the other hand, the more native forest cover, the lower the water yield (Bayer, 2014; Molin, 2014; Burt & Swank, 1992). Thus, the progressive increase of water yield from 2010 to 2035, compared to a higher total forest cover, could indicate other factors, such as forest connectivity, forest climax and secondary factors such as BMP, that could produce non-linear conditions of water yield from the local scale to the catchment scale.

Likewise, water yield is related to the absolute value of integrating the flow duration curve. For example, the flow duration curve of S1 (1990) exceeded other scenario curves in approximately 75% of time, with differentiated behaviour in both peak flows (lower probability) and low flows (higher probability of duration curves).

### 3.5. Results of a new index for hydrologic service assessment

A new index for hydrologic service assessment was developed as a simple relation between greyWF and water yield, using a fraction between water demand (numerator) and availability (denominator). Some authors commonly use this fraction as a direct approach to water scarcity (i.e. Smakhtin, et al., 2005; Hoekstra et al,





.2013; McNulty et al., 2010; among others). Therefore, we first assessed greyWF by respective drainage basins
(**Figure 15**). Then, we calculated the water pollution levels.
Results in **Figure 16** show the composite water pollution level (WPLcomposite) *versus* drainage areas and
compared with the HSI. The baseline *WPLcomposite,ref* is related to the *Domithildes* catchment (horizontal,
dotted line in **Figure 16**). This line divides the graph into two regions: less sustainable basins (*HSI*>0) and more
sustainable basins (*HIS*<=0). More sustainable basins (*HIS*<0) are *Salto, Cachoeira* nested catchments
(*Cachoeira dos Pretos, Chalé Ponto Verde* and *Ponte Cachoe*ira), as well as *F28, F24* and the *Upper Jaguari*
basin.

### 616     3.6. Comparison of field investigation and modelled scenarios

**Figure 17** compares field, experimental data (Taffarello et al., 2016a) with modelled scenarios of land-use and
land-cover change, including the EbA hypothesis. The horizontal axis of **Figure 17** depicts the water yield of
each scenario or water security condition, for disaster risk reduction with EbA. Reference flows were assessed
from official policy institutions (see DAEE, 1987).

### 621     4. Conclusions and Recommendations

Although the water-forest system interaction is a classic issue in Hydrology (Hibbert, 1967; Tucci & Clarke,
1998; Adreássian, 2004; Zhao et al., 2012), the impacts of vegetation on quali-quantitative aspects of water
resources need to be better understood.
Supported by field experiments and quali-quantitative simulations under different scenarios including EbA
options with BMP, our results showed evidence of nonlinear relationships among LULC, water yield, greyWF of
nitrate, total phosphorus and sediments, which irreversibly affect the composite of water pollution level (WPL),
the definition of WPL of reference (here established at Domithildes catchment) and the hydrologic service index
(HSI). Despite using a semi-distributed model for assessing non-point sources of pollution mainly tested under
different LULC scenarios, our results showed that the intrinsic nature of flow-load duration curves, LULC and
greyWF are constrained to high uncertainties and nonlinearities both from *in-situ* sampling and from processes
interactions of modelling. Our results show the need to evaluate many uncertainty sources, such as: model
sensitivity analysis, observed streamflow data, ecohydrologic model performance, residual analysis, etc. To
attain goals of EbA, using HSI through greyWF assessment and composite of WPL, some conditions are needed,
as follows: (i) to avoid the inputs of high-concentrated pollutants, especially growing urban settlements, (ii) to
restore riparian vegetation and (iii) trapping and removing inflowing sediments. For the health of river
ecosystems, we used HSI, flow regimes and WPLcomposite, as an alternative proposal to define environmental
flows (Tharme, 2003; Olden et al., 2011; Poff & Zimmerman, 2010; Poff & Matthews, 2013). Although the role
of vegetation on streamflow has been widely studied, very few investigations have been reported in Brazil with
control nutrient sources, transportation and delivery. Moreover, further field and modelling research is needed



when integrating LULC, EbA and greyWF. Thus, this future research could clarify the influence of vegetation on
water quality and the role of anthropogenic and natural drivers in ecohydrologic processes on a catchment-scale.
**5. Acknowledgments**
This study was supported by the Sao Paulo Research Foundation (FAPESP) [grants #2012/22013-4;
#2014/15080-2; and #2008/58161-1 "Assessment of Impacts and Vulnerability to Climate Change in Brazil and
Strategies for Adaptation Options"], CAPES 88887.091743/2014-01 (ProAlertas CEPED/USP), CNPq
465501/2014-1 & FAPESP 2014/50848-9 INCT-II (Climate Change, Water Security), CNPq PQ 312056/2018-8
(EESC-USPCEMADEN/MCTIC) & CAPES PROEX (PPGSHS, EESC/USP).We thank two graduates in
environmental engineering at USP-Lorena, Cauê Fontão and Rodolfo Cursino, for providing updated
information on water footprint for the introduction of the manuscript.

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





TABLES

**Table 1:** Sub-basins delimited in SWAT with drainage areas and geographic locations.

| SWAT sub-basin | Gauge station | Drainage area (km²) | Coordinates Lat. | Long. |
|---|---|---|---|---|
| 1 | AltoJaguari | 302.2 | -22.820 | -46.154 |
| 2 | F23 | 508.1 | -22.827 | -46.314 |
| 3 | F28 | 276.8 | -22.806 | -45.989 |
| 4 | Salto | 15.0 | -22.838 | -46.218 |
| 5 | Pq Eventos | 926.5 | -22.853 | -46.325 |
| 6 | Posses Exut | 11.9 | -22.833 | -46.231 |
| 7 | Portal das Estrelas | 7.1 | -22.820 | -46.244 |
| 8 | F25B | 971.9 | -22.850 | -46.346 |
| 9 | Domithildes | 9.9 | -22.886 | -46.222 |
| 10 | B: Jaguari | 1037.0 | -22.896 | -46.385 |
| 11 | F30 | 15.1 | -22.935 | -46.212 |
| 12 | Ponte Cach. | 121.0 | -22.967 | -46.171 |
| 13 | Chale Pt Verde | 107.9 | -22.964 | -46.181 |
| 14 | Cach Pretos | 101.2 | -22.968 | -46.171 |
| 15 | B: Jacarei | 200.5 | -22.959 | -46.341 |
| 16 | F24 | 293.5 | -22.983 | -46.244 |
| 17 | B: Cachoeira | 391.7 | -46.209 | -46.276 |
| 18 | F34 | 129.2 | -23.073 | -46.209 |
| 19 | B: Atibainha | 313.8 | -23.182 | -46.342 |
| 20 | Moinho | 16.9 | -23.209 | -46.357 |

**Table 2:** Characteristics of quantitative calibration and validation of SWAT in studied catchments (Moriasi et al., 2007):

| Gauge station | Area (km²) | Pbias (%) | NSE (-) | NSE Log (-) | Pbias (%) | NSE (-) | NSE Log(-) | Performance level of calibration and validation (Moriasi et al., 2007) |
|---|---|---|---|---|---|---|---|---|
| | | Calibration | | | Validation | | | |
| Posses | 13.3 | -22.0 | 0.68 | 0.52 | 15.4 | 0.78 | 0.38 | Unsatisfactory/very good |
| F28 | 281.5 | 5.3 | 0.80 | 0.68 | 14.2 | 0.72 | 0.31 | Very good/good |
| F24 | 294.5 | -13.3 | 0.69 | 0.71 | -1.7 | 0.65 | 0.34 | Satisfactory/satisfactory |
| Atibainha | 331.7 | -14.5 | 0.60 | 0.55 | 1.7 | 0.71 | 0.54 | Satisfactory/good |
| Cachoeira | 397.3 | -26.6 | 0.49 | 0.31 | -46.7 | 0.27 | 0.05 | Unsatisfactory/unsatisfactory |
| F23 | 511.2 | -1.8 | 0.88 | 0.90 | 12.0 | 0.84 | 0.77 | Very good/ very good |
| F25B | 981.4 | 3.6 | 0.91 | 0.89 | 11.4 | 0.77 | 0.72 | Very good/ very good |
| Jag+Jac | 1276.9 | -12.0 | 0.83 | 0.87 | -8.4 | 0.82 | 0.73 | Very good/ very good |



**Table 3:** Calibrated SWAT parameters in the headwaters of the Cantareira Water Supply System.

| | Description | Parameter | Fitted values |
|---|---|---|---|
| Water Quantity | Initial SCS curve number (moisture condition II) for runoff potential. | CN2 | <0.25 |
| | Soil evaporation compensation factor. | ESCO | <0.2 |
| | Plant uptake compensation factor. | EPCO | <1.0 |
| | Maximum canopy storage (mm). | CANMX | Varies by vegetal cover |
| | Manning's coefficient "n" value for the main channel. | CH_N2 | 0.025 |
| Water Quality | Nitrate percolation coefficient | NPERCO | 0.2 |
| | Minimum value of the USLE C coefficient for water erosion related to the land cover | USLE_C | Varies by land use (< 0.4) |





**Table 4:** Maximum and minimum values of quali-quantitatives variables observed during field campaigns of Oct, 2013 - May, 2014 in the headwaters of the Cantareira System, Southeast Brazil.

| Sub-basin | Flow discharge | | Electrical conductivity | | pH | | BOD | | COD | | E: Coli | |
|---|---|---|---|---|---|---|---|---|---|---|---|---|
| | MIN: (m³/s) | MAX: (m³/s) | MIN (µS/cm) | MAX (µS/cm) | MIN. | MAX. | MIN (mg./L) | MAX (mg/.L) | MIN (mg/.L) | MAX (mg./L) | MIN (ufc) | MAX (ufc) |
| Upper Posses | 0,009 | 0,034 | 54 | 63 | 6,6 | 7,0 | <1 | <1 | 6 | 19 | 10 | 870 |
| Middle Posses | 0,031 | 0,082 | 53 | 63 | 6,8 | 7,0 | <1 | <1 | 8 | 26 | 14 | 260 |
| Outlet Posses | 0,039 | 0,107 | 65 | 133 | 6,7 | 7,1 | 2 | 2 | 5 | 24 | 1 | 2000 |
| Outlet Salto | 0,032 | 0,093 | 22 | 62 | 6,6 | 7,2 | 4 | 4 | 4 | 22 | 4 | 4800 |
| F23 | 1,706 | 5,500 | 44 | 60 | 6,7 | 6,9 | 6 | 6 | 18 | 48 | 17 | 3600 |
| Upper Jaguari | 1,387 | 6,283 | 23 | 59 | 6,9 | 7,0 | 2 | 2 | 2 | 28 | 2 | 100 |
| Parque de Eventos | 4,568 | 20,689 | 38 | 50 | 6,6 | 6,9 | 2 | 6 | 11 | 36 | 31 | 4100 |
| Cachoeira dos Pretos | 1,460 | 3,060 | 13 | 17 | 6,7 | 7,0 | <1 | <1 | 6 | 20 | 33 | 37 |
| Chalé Ponto Verde | 1,540 | 3,223 | 14 | 16 | 6,8 | 7,1 | <1 | 2 | 6 | 21 | 3 | 290 |
| Ponte Cachoeira | 1,400 | 3,618 | 15 | 20 | 6,3 | 7,0 | 2 | 3 | 6 | 26 | 340 | 4000 |
| F24 | 2,250 | 5,174 | 22 | 28 | 6,7 | 6,9 | 2 | 4 | 10 | 34 | 5 | 690 |
| Intervention Cancã | 0,005 | 0,022 | 39 | 48 | 6,7 | 7,0 | 3 | 3 | 3 | 22 | 40 | 730 |
| Reference Cancã | 0,002 | 0,009 | 42 | 48 | 6,6 | 7,1 | 2 | 2 | 5 | 27 | 5 | 650 |
| F30 | 0,641 | 1,297 | 36 | 40 | 6,8 | 7,1 | 3 | 4 | 9 | 42 | 140 | 3400 |
| Intervention Moinho | 0,003 | 0,055 | 34 | 41 | 6,1 | 7,1 | 5 | 8 | 6 | 22 | 17 | 160 |
| Reference Moinho | 0,004 | 0,017 | 34 | 35 | 6,7 | 6,9 | <1 | <1 | 4 | 16 | 690 | 2400 |
| Outlet Moinho | 0,081 | 0,162 | 51 | 60 | 6,8 | 7,0 | <1 | <1 | 6 | 23 | 99 | 1300 |



**Table 5:** LULC changes in 20 sub-basins, headwaters of the Cantareira System for scenarios of S1 (LULC in 1990), S2 (LULC in 2010) and S2+EbA (LULC in 2035).

| Sub-basin | Gauge station | Dranaige area(km²) | Equivalent scenario timeline | Native forest | Euca-lypto | Pasture | Agri-culture | Urban |
|---|---|---|---|---|---|---|---|---|
| | | | | Land-Use/Land-Cover (% of drainage area) | | | | |
| 1 | Upper Jaguari | 302.20 | 1990 | 47 | 6 | 35 | 12 | 0 |
| | | | 2010 | 33 | 13 | 34 | 20 | 0 |
| | | | 2035 | 66.2 | 21.1 | 8.2 | 4.6 | 0.3 |
| 2 | F23 | 508.10 | 1990 | 37 | 2 | 52 | 9 | 0 |
| | | | 2010 | 34 | 2 | 44 | 19 | 0 |
| | | | 2035 | 36.2 | 2.3 | 42.5 | 18.6 | 0.5 |
| 3 | F28 | 276.80 | 1990 | 78 | 8 | 11 | 3 | 0 |
| | | | 2010 | 69 | 22 | 6 | 3 | 0 |
| | | | 2035 | 69.1 | 21.3 | 6 | 3.3 | 0.3 |
| 4 | Salto | 15.06 | 1990 | 40 | 1 | 50 | 9 | 0 |
| | | | 2010 | 29 | 2 | 53 | 16 | 0 |
| | | | 2035 | 31.5 | 2.4 | 50.5 | 15.5 | 0 |
| 5 | Pq: Eventos | 926.50 | 1990 | 35 | 1 | 50 | 11 | 3 |
| | | | 2010 | 36 | 2 | 44 | 15 | 3 |
| | | | 2035 | 45.8 | 8.2 | 31.9 | 13.5 | 0.6 |
| 6 | Posses outlet | 11.99 | 1990 | 22 | 2 | 67 | 9 | 0 |
| | | | 2010 | 13 | 1 | 70 | 16 | 0 |
| | | | 2035 | 15.6 | 0.7 | 70.2 | 13.5 | 0 |
| 7 | Portal Estrelas | 7.17 | 1990 | 24 | 0 | 62 | 14 | 0 |
| | | | 2010 | 15 | 1 | 72 | 12 | 0 |
| | | | 2035 | 17.1 | 0.6 | 70.5 | 11.8 | 0 |
| 8 | F25B | 971.90 | 1990 | 33 | 2 | 50 | 10 | 5 |
| | | | 2010 | 38 | 1 | 43 | 13 | 5 |
| | | | 2035 | 45.5 | 7.9 | 32.3 | 13.5 | 0.8 |
| 9 | Domithildes | 9.93 | 1990 | 51 | 0 | 37 | 12 | 0 |
| | | | 2010 | 52 | 5 | 30 | 13 | 0 |
| | | | 2035 | 56.4 | 4.6 | 27.3 | 11.7 | 0 |
| 10 | B: Jaguari* | 1037.00 | 1990 | 37 | 1 | 52 | 11 | 0 |
| | | | 2010 | 40 | 2 | 41 | 16 | 0 |
| | | | 2035 | 45 | 8 | 32.6 | 13.6 | 0.8 |
| 11 | F30 | 15.14 | 1990 | 30 | 1 | 57 | 12 | 0 |
| | | | 2010 | 28 | 4 | 54 | 14 | 0 |
| | | | 2035 | 47.3 | 4.4 | 35,8 | 12.5 | 0 |
| 12 | Ponte Cachoeira | 121.00 | 1990 | 31 | 0 | 62 | 7 | 0 |
| | | | 2010 | 31 | 9 | 48 | 11 | 0 |
| | | | 2035 | 58.9 | 20.1 | 15.3 | 5.7 | 0 |
| 13 | Chale Pt: Verde | 107.90 | 1990 | 39 | 8 | 46 | 7 | 0 |
| | | | 2010 | 29 | 31 | 30 | 10 | 0 |
| | | | 2035 | 62,1 | 21.5 | 11 | 5.1 | 0 |
| 14 | Cachoeira dos Pretos | 101.20 | 1990 | 59 | 8 | 27 | 6 | 0 |
| | | | 2010 | 66 | 20 | 9 | 5 | 0 |
| | | | 2035 | 66.2 | 20.3 | 8.7 | 4.6 | 0 |
| 15 | B: Jacareí* | 200.50 | 1990 | 32 | 0 | 52 | 13 | 2 |
| | | | 2010 | 39 | 5 | 42 | 13 | 2 |
| | | | 2035 | 32.7 | 2.7 | 32.1 | 10.3 | 2 |
| 16 | F24 | 293.50 | 1990 | 56 | 4 | 32 | 8 | 0 |
| | | | 2010 | 47 | 18 | 25 | 9 | 0 |
| | | | 2035 | 53.2 | 17.8 | 21.3 | 7.7 | 0 |
| 17 | B: Cachoeira* | 391.70 | 1990 | 35 | 6 | 47 | 11 | 0 |
| | | | 2010 | 42 | 21 | 27 | 10 | 0 |
| | | | 2035 | 50.1 | 18.1 | 22 | 7.9 | 0 |
| 18 | F34 | 129.20 | 1990 | 59 | 9 | 23 | 9 | 0 |
| | | | 2010 | 61 | 19 | 10 | 10 | 0 |
| | | | 2035 | 61.4 | 19.3 | 9.9 | 9.3 | 0 |
| 19 | B.Atibainha* | 313.80 | 1990 | 49 | 7 | 30 | 13 | 0 |
| | | | 2010 | 60 | 18 | 13 | 9 | 0 |
| | | | 2035 | 56.3 | 17.5 | 10.8 | 8.8 | 0 |
| 20 | Moinho | 16.90 | 1990 | 46 | 10 | 27 | 17 | 0 |
| | | | 2010 | 49 | 22 | 17 | 13 | 0 |
| | | | 2035 | 49.9 | 21.4 | 16.2 | 12.5 | 0 |





FIGURES

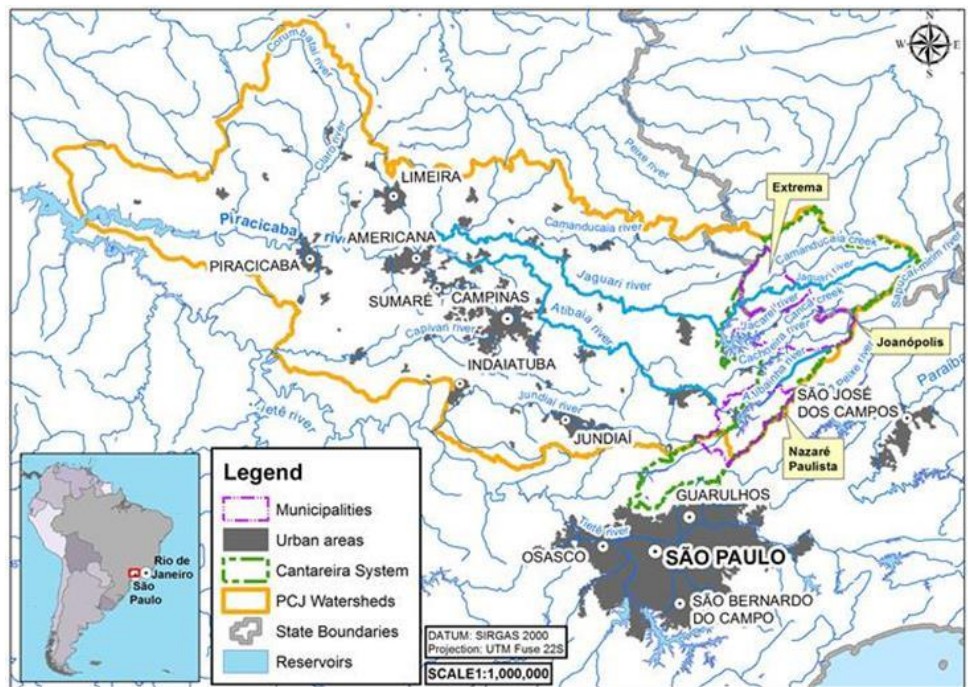

**Figure 1:** Location of Cantareira Water Supply System in the Piracicaba and Upper Tietê watersheds.




**Figure 2:** Methodological scheme for assessing hydrologic services based on greyWF.





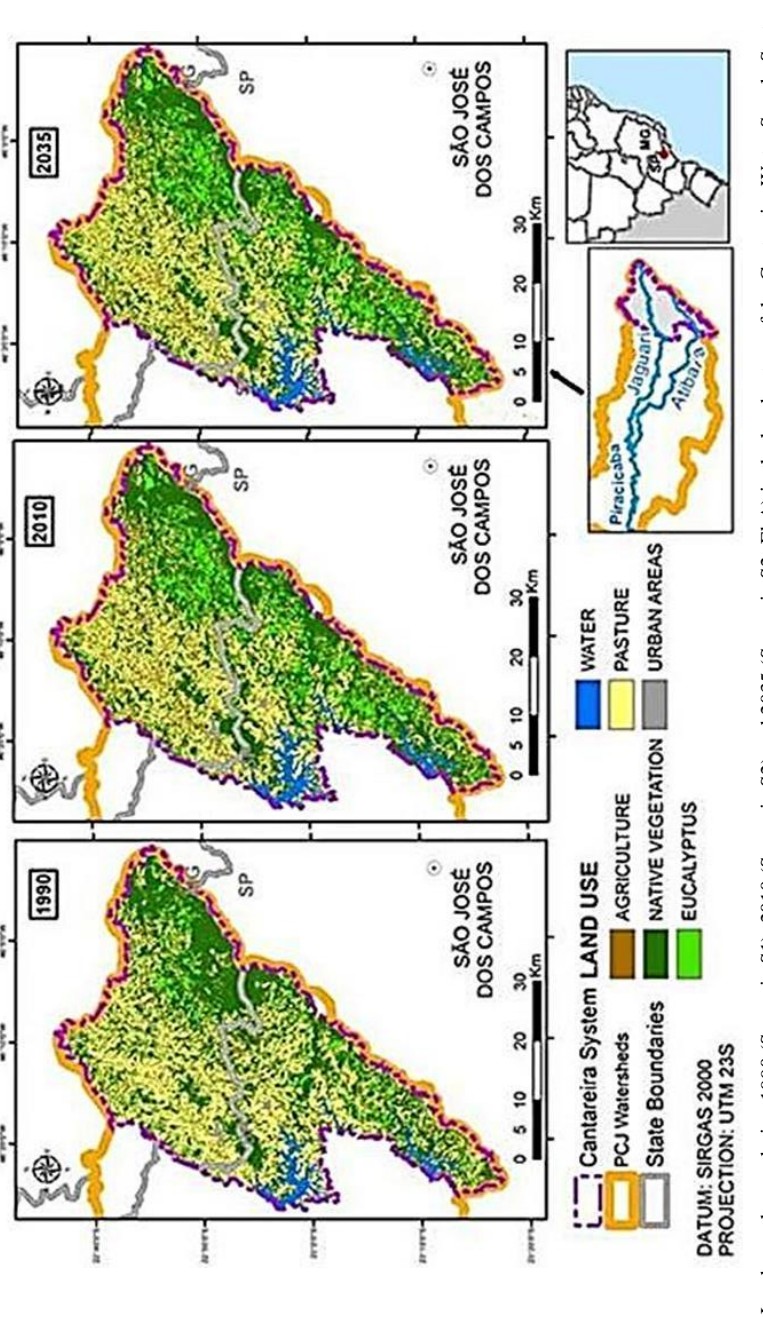

**Figure 3:** Land-use change during 1990 (Scenario S1), 2010 (Scenario S2) and 2035 (Scenario S2+EbA) in the headwaters of the Cantareira Water Supply System:



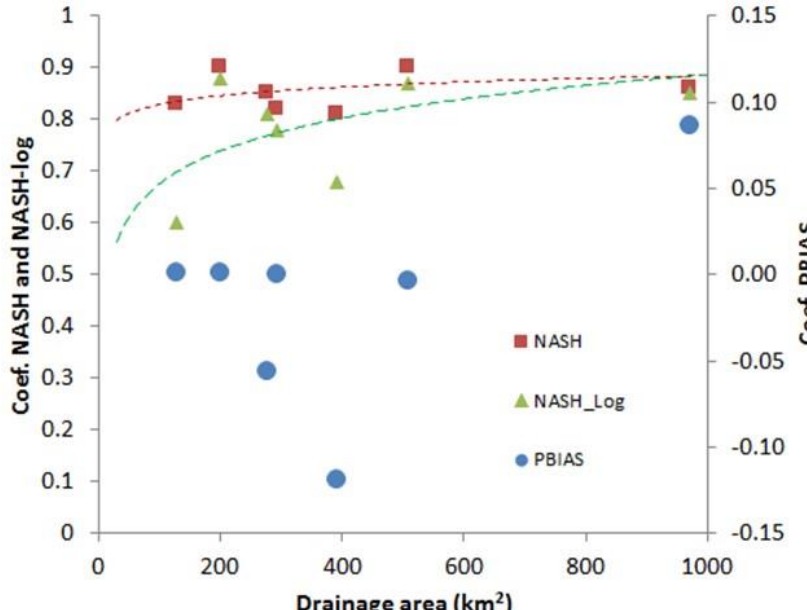

**Figure 4:** Model calibration related to drainage areas of catchments in the Cantareira System.





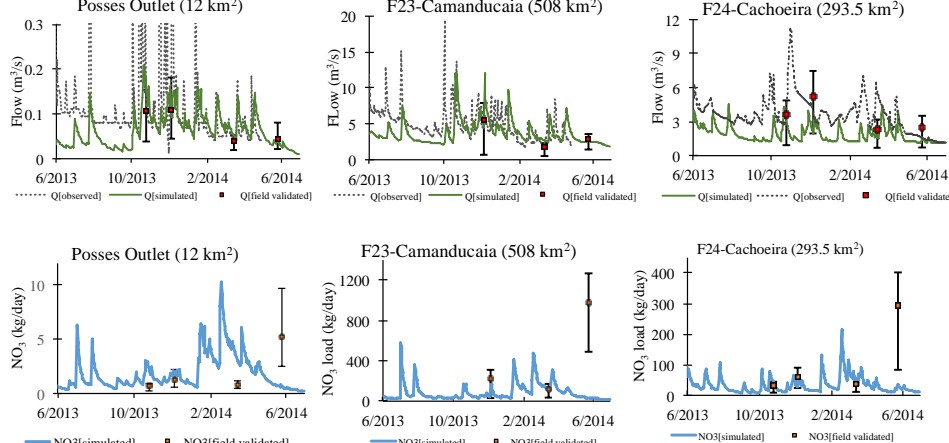

**Figure 5:** Comparison between flow discharges (upper part) and nitrate loads (lower part), through observed (dotted lines), simulated by SWAT (solid lines) and field validation through instantaneous experimental samples (marked points with uncertainty intervals) at monitored stations of *Posses Outlet* (left part), *F23 Camanducai*a (center part) and *F24-Cachoeira* (right part). Time (horizontal axis) is represented by month/year. The uncertainty bars were determined using instantaneous velocities measured in the river cross-sections during 2013/14 field campaigns (see Taffarello et al, 2016-a).





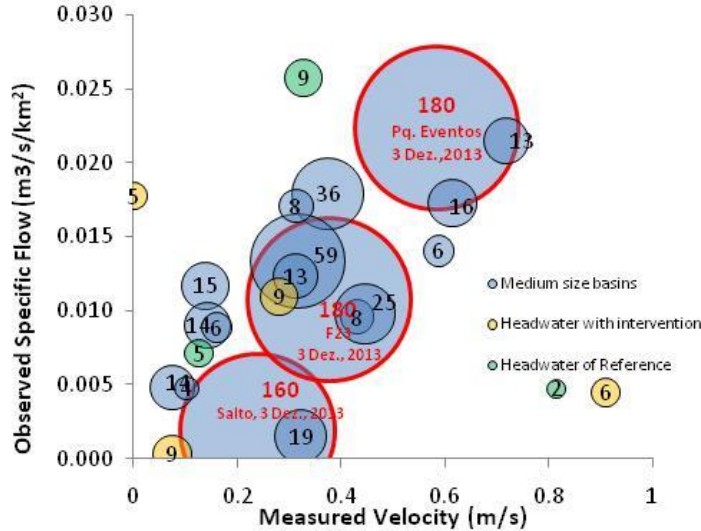

**Figure 6:** Experimental sampling of turbidity (size of circles), observed flows and mean velocities in river cross sections of 17 catchments in Cantareira System headwater (Oct, 2013 - May, 2014).

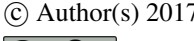



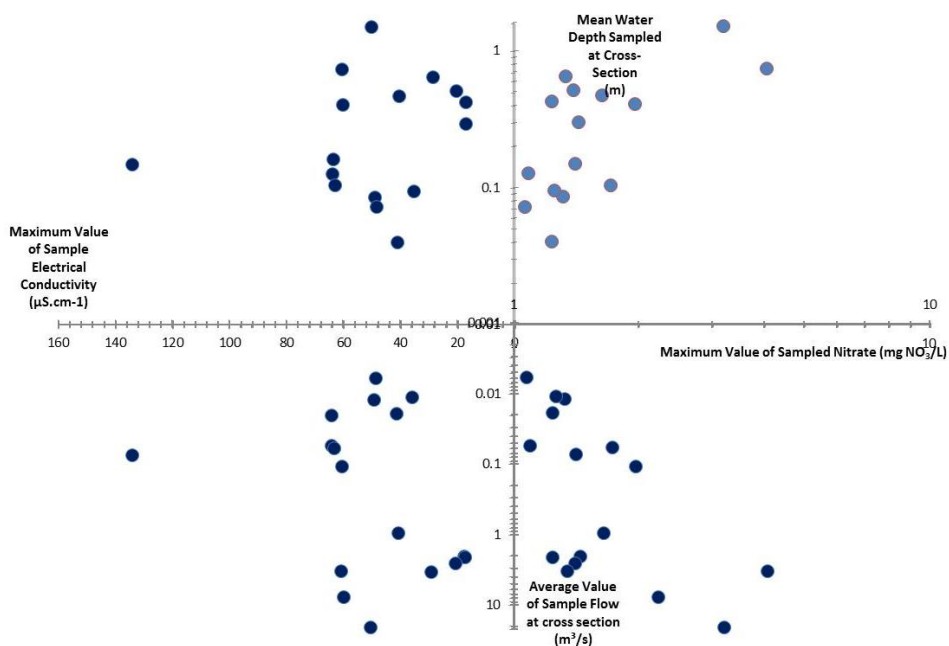

**Figure 7:** Multidimensional chart of hydraulic and water quality variables sampled in field campaigns in the headwaters of the Cantareira Water Supply System between Oct, 2013 - May, 2014.



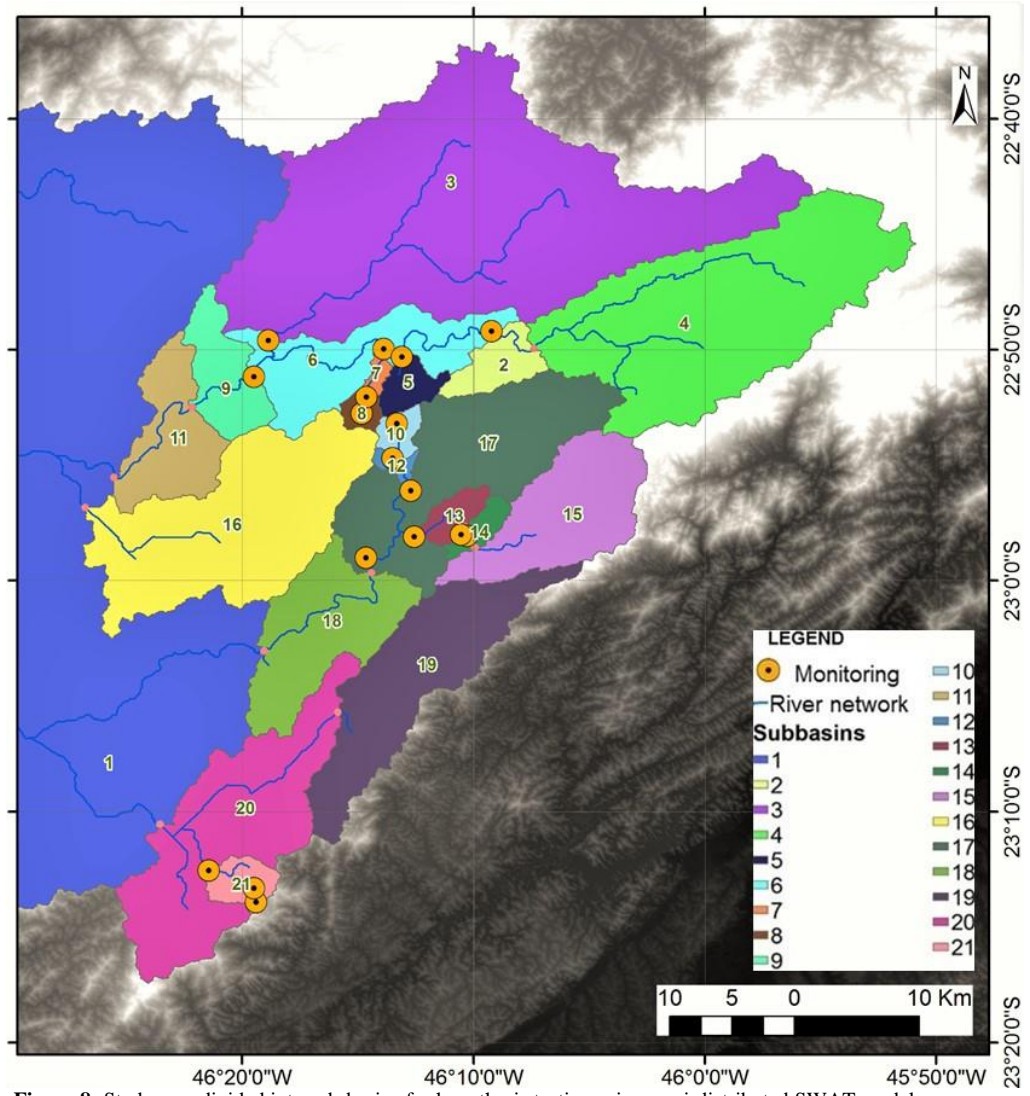

**Figure 8:** Study area divided into sub-basins for hypothesis testing using semi-distributed SWAT model.

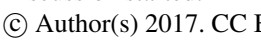


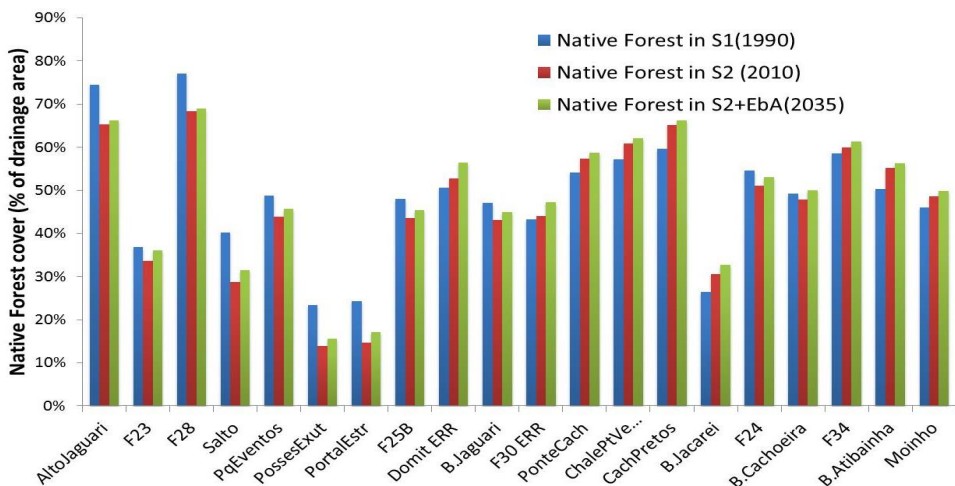

**Figure 9:** Native forest cover in S1 (1990), S2 (2010) and S2+EbA (2035).



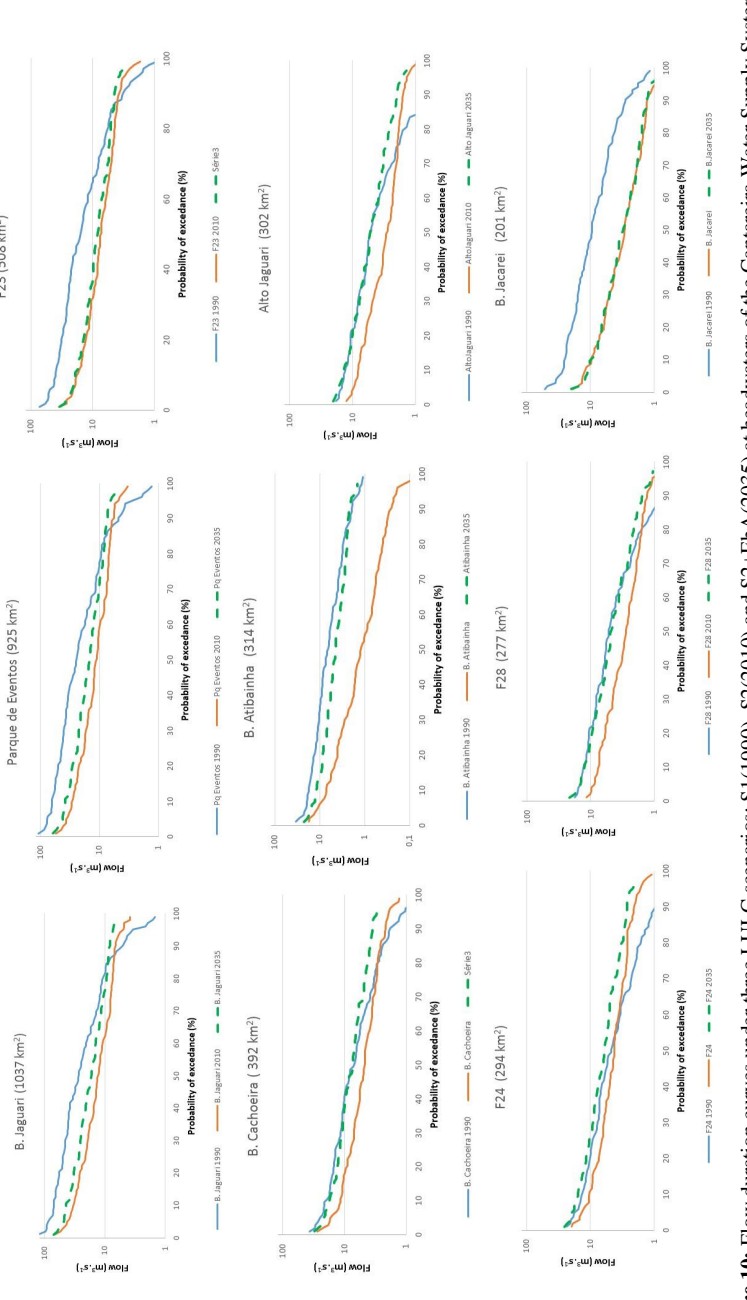

**Figure 10:** Flow duration curves under three LULC scenarios: S1(1990), S2(2010) and S2+EbA(2035) at headwaters of the Cantareira Water Supply System.


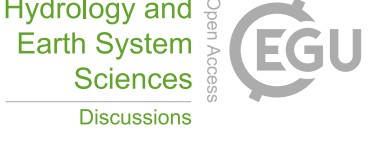

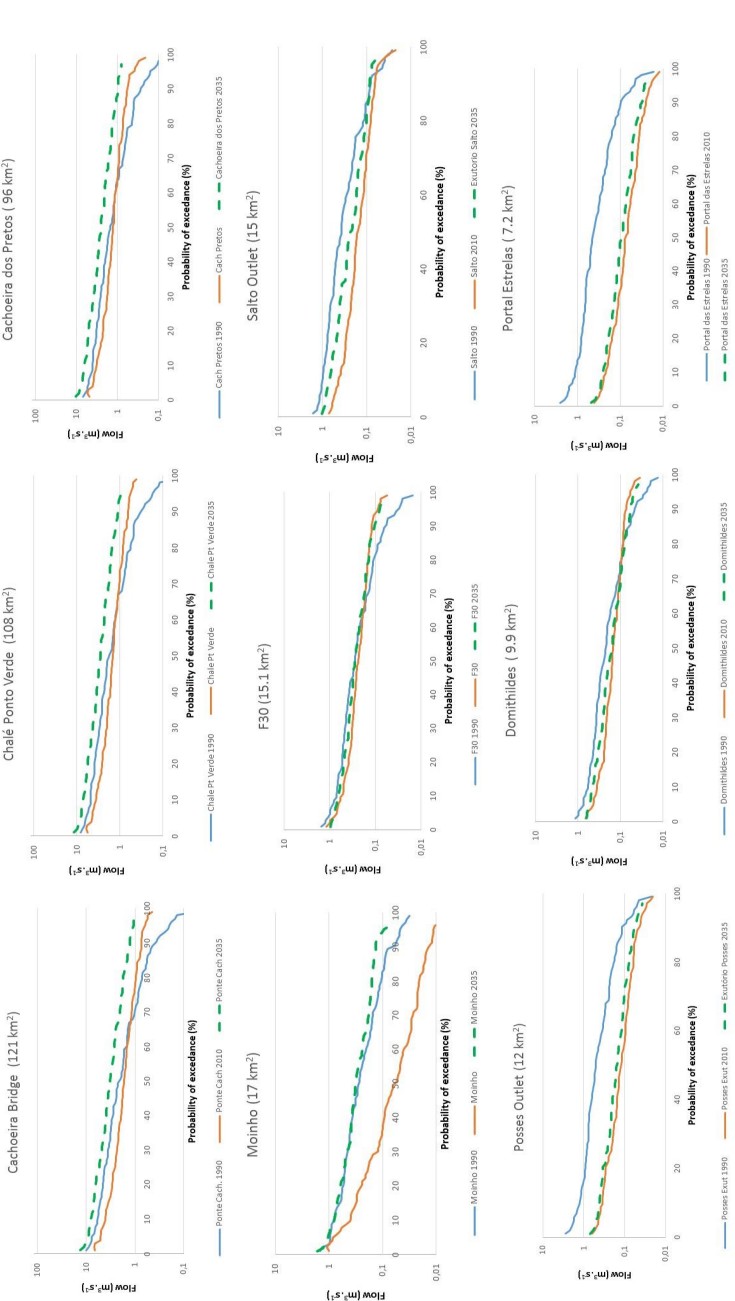

**Figure 10:** Flow duration curves under three LULC scenarios: S1(1990), S2(2010) and S2+EbA(2035) at headwaters of the Cantareira Water Supply System(cont.).




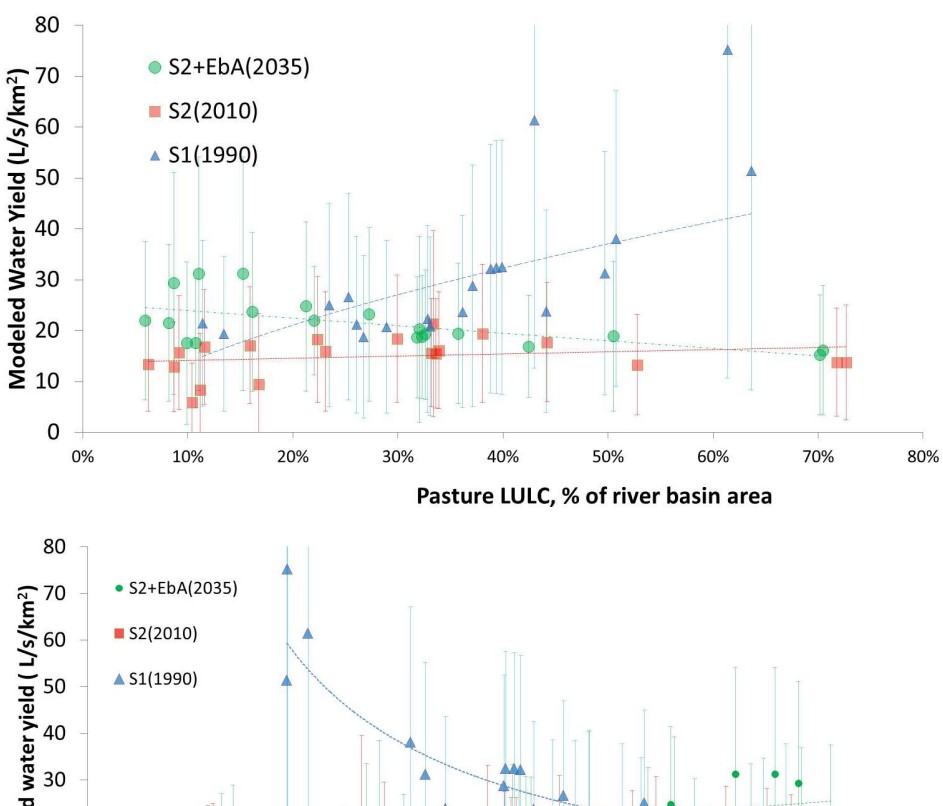

**Figure 11:** LULC scenarios for specific water yield for 20 drainage areas at Jaguari, Cachoeira and Atibainha watersheds, according to S1 (1990), S2 (2010) and S2+EbA (2035) scenarios.



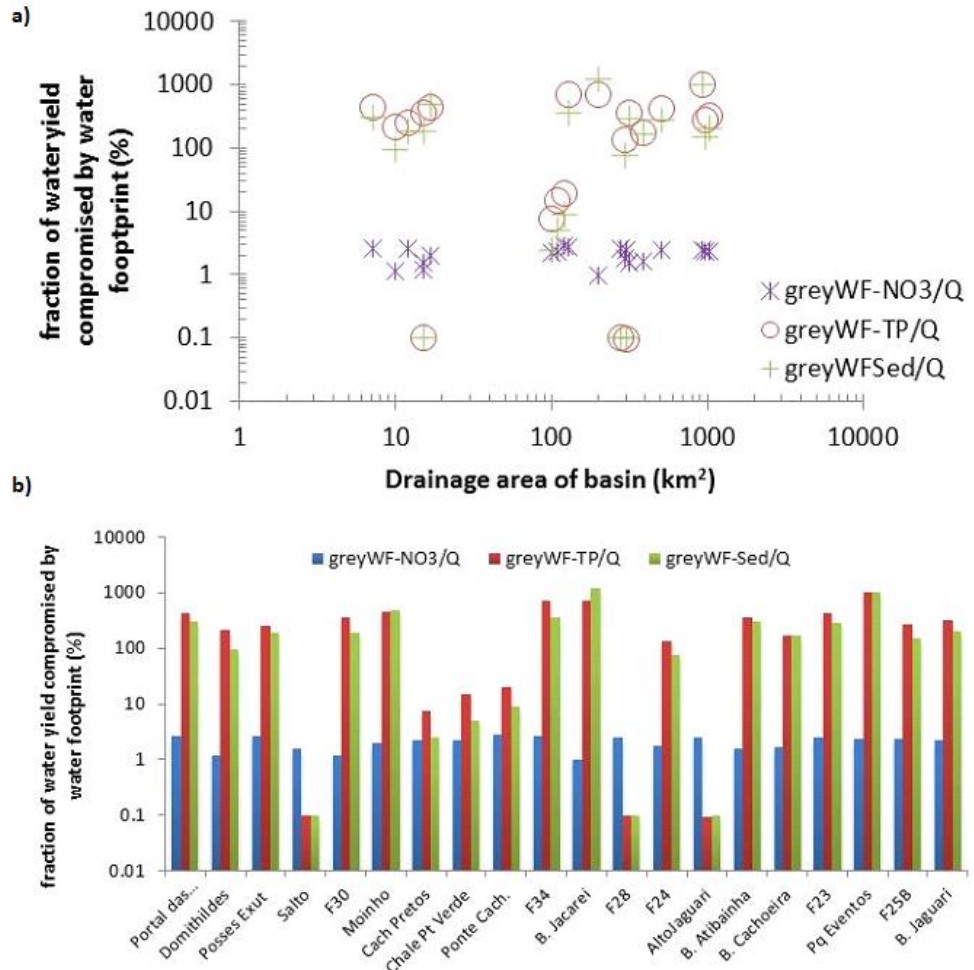

**Figure 12:** Fraction of water yield compromised by the grey water footprint for nitrate, total phosphorous and sediments versus drainage area **(a)**, and showing the studied subbasins **(b)**.



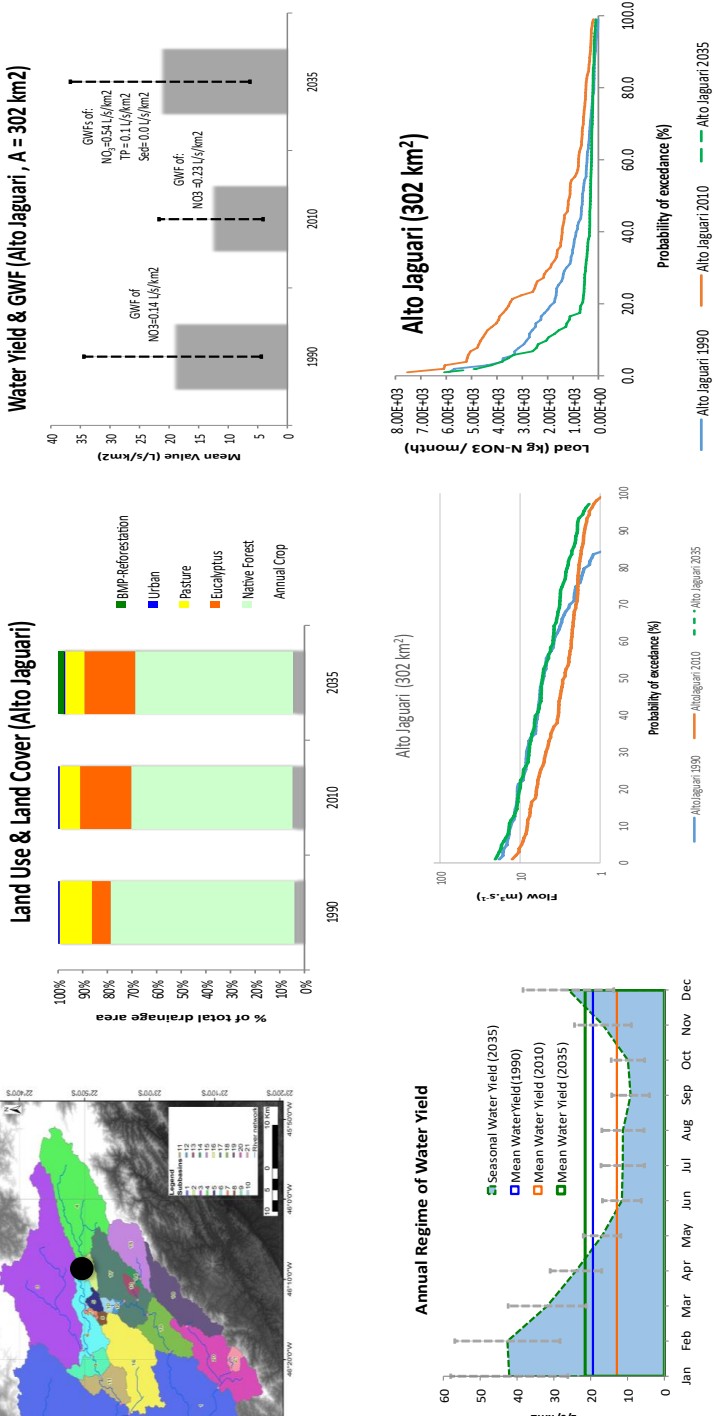

**Figure 13:** Synthesis chart of case study *Upper Jaguari* sub-basin (drainage area = 302 km²). Left, upper chart: localization at the drainage areas of Cantareira System: Center, upper chart: LULC conditions for scenarios S1 (1990), S2 (2010) and S2+EbA (2035): Right, upper chart: comparison of water yields simulated for conditions of S1, S2 and S2+EbA: Left, lower chart: water yield scenarios compared with intra-annual regime of S2+EbA scenario: Center, lower chart: comparison of duration curves of flows for S1, S2 and S2+EbA conditions: Right, lower chart: duration curves of N-NO3 loads for S1, S2 and S2+Eb.A




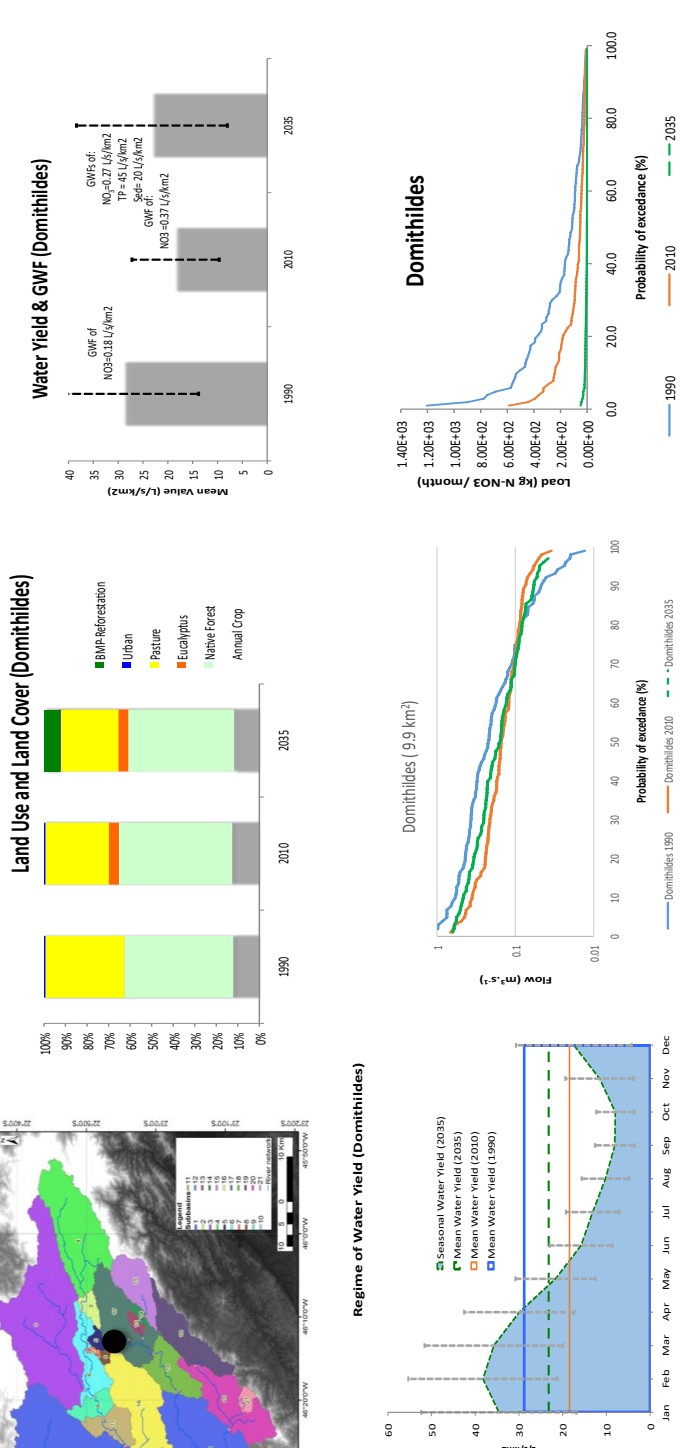

**Figure 14:** Synthesis chart of case study *Domithildes* catchment (drainage area = 9.9 km²). Left, upper chart: localization at the drainage areas of the Cantareira System: Center, upper chart: LULC conditions for scenarios S1 (1990), S2 (2010) and S2+EbA (2035): Right, upper chart: comparison of water yields simulated for conditions of S1, S2 and S2+EbA: Left, lower chart: water yield scenarios compared with intra-annual regime of S2+EbA scenario: Center, lower chart: comparison of duration curves of flows for S1, S2 and S2+EbA conditions: Right, lower chart: duration curves of N-NO3 loads for S1, S2 and S2+EbA.



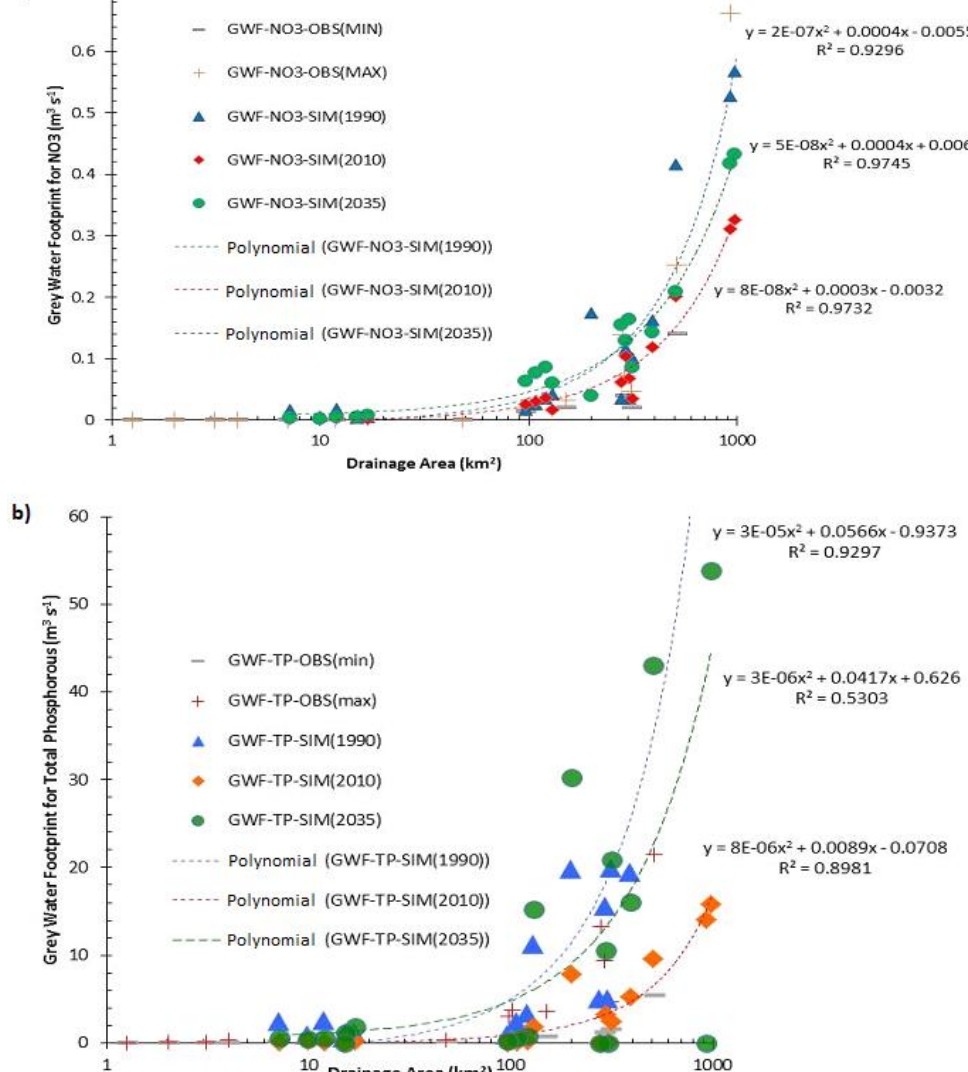

**Figure 15:** Relationships between Grey Water Footprint for Nitrate (a) and Total Phosphorous (b) according to three LULC scenarios (1990, 2010 and 2035) and size of the drainage areas of headwaters in the Cantareira Water Supply System.





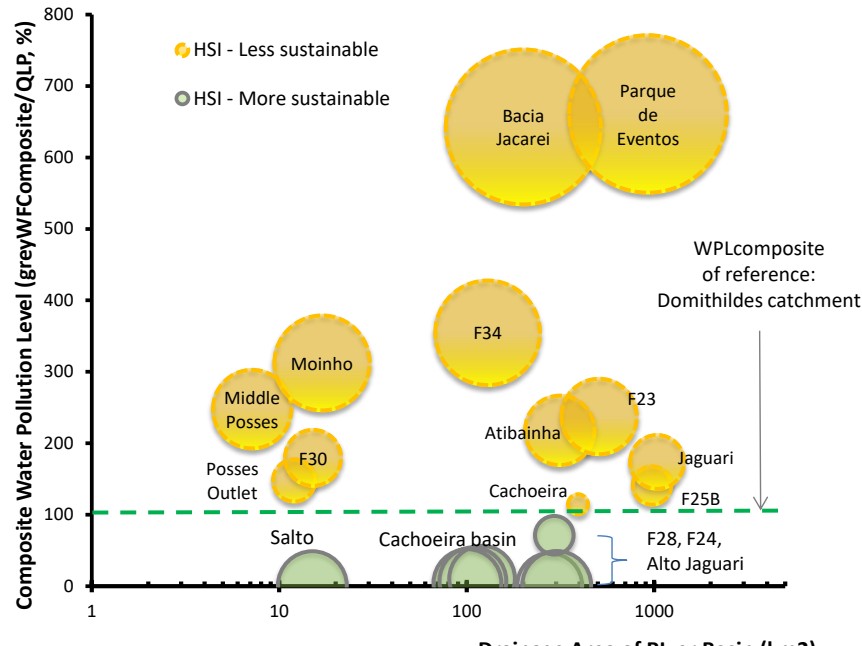

**Figure 16:** Hydrologic Service Index (circle ratio) related to drainage area of river basin (horizontal axis) and composite of water pollution index (vertical axis) for S2+EbA scenario: Equal weights of nitrate, total phosphorus and dissolved sediments are expressed in *WPLcomposite*.







**Figure 17:** Summary of monitored and modelled water yield (horizontal axis), compared with ecosystem-based adaptation and grey water footprint in the headwaters of the Cantareira System, Brazil. Upper bars represent modelling freshwater quality scenarios ("blue": S1, 1990; "orange": S2, 2010; and "green": S2+Eba, 2035). Middle red bars depict regionalized long-term water yield (Qlp) and reference flows of duration curves (Q90% and Q95%) regarding Brazilian regulatory agencies (DAEE, 1988). Lower blue bars depict monitored water yield in several catchments of Cantareira System during the 2013/14 drought (see Taffarello et al, 2016-a). Intervals of greyWF of scenarios are also plotted. Bold, capital letters ("A", "B", "C", "D", "E"), showing different conditions for water security using deviations from regionalized long-term water yield (Qlp) for the headwaters of Cantareira System, Brazil.

