# Peer review of "Modelling freshwater quality scenarios with ecosystem-based adaptation in the headwaters of the Cantareira System, Brazil"

_Hydrology and Earth System Sciences, 2017_

## Referee Comment (RC1) · Anonymous Referee #1 · 7 Oct 2017

Review report for Manuscript ID HESS-2017-474 entitled "Modelling freshwater quality scenarios with ecosystem-based adaptation in the headwaters of the Cantareira system, Brazil"

General comment The paper compares freshwater quality scenarios under different land-use/land cover changes in the headwaters of the Cantareira system, Brazil. Soil and Water Assessment Tool (SWAT) is used to model water yield, nitrate and total phosphorus loads, and sediment yields. The Hydrological Service Index is developed for 20 sub-basins by considering the grey water footprint for nitrate, total phosphorus and sediments yield in order to assess the sustainability of the hydrological services.

[Figure]

The study reported restoration of forest cover conversion scenario through ecosystem-based adaptation in protected areas foreseen additional best management practices at the headwaters of the water supply systems. The paper is interesting and suitable for publication after major revisions. The main changes should be done in how the SWAT model is representing the study area especially during drought years. One of the main reasons for the discrepancies between monitoring data/existing literatures and model simulations might be the weakness of SWAT model to capture extreme flows or water yields. Besides, how the model parameters are selected for the calibration and validation of SWAT model. Improve the Tables and Figures to be more informative to the reader (Please see on specific comments part below). The specific and thechnical comments are as follows: The abstract section could be concise. 1. Line 55 (Colombia, 2015, 2014, 2010). 2. Lines 58 to 66 "Hoekstra et al., 2011" is over cited. Could be rephrased. 3. Line 141 run from 2009 to 2014. 4.Line 153 "three data collection platforms" their geographic locations could be indicated on the study area map. 5. Line 156 the type of secondary data could be clearly indicated. 6. Lines 252-255. Besides adopting from the existing literatures, implementing sensitivity analysis could be recommended in order to select model parameters. 7. Lines 276-279. It is known that SWAT model is not for extreme flows and hence water quality parameters. 8. Line 299 could be moved to line 298. 9. Line 310 could be moved to line 309. 10. Line 322 could be moved to line 321. 11. Line 455 to 456 should be written with appropriate multiplication sign. 12. Line 514. It would be useful to relate spatially the sub-basins in which the differences in land-use/land-cover are the greatest and the water yield, nitrate, total phosphorus and sediments yield differences are evident. For instance providing maps which indicate temporal changes in LULC and corresponding changes in water quality parameters considered. 13. Line 533. Reason for selecting the two sub-basins among the 20 sub-catchments? 14. Lines 535. Any statistical relationship between the changes in LULC classes and grey water footprints. For instance multivariate statistical analysis. 15. Lines 544 to 555. As one-third of the SWAT simulation are low-flow or drought years. It is known that SWAT model is weak in capturing extreme flows.

One of the reasons for the discrepancy between monitoring data and model simulation might not the weakness of the SWAT model to represent low-flows? 16. Table 1. It might be better to replace sub-basin coordinates with key modelling results and/or field observations. 17. Table 2. Possible reason for model underperformance for some sub-basins? 18. Table 3. The selected SWAT parameters are not exhaustive unless sensitivity analysis is conducted. 19. Table 5. I would like to see additional column indicating the Hydrologic Services Index. The symbol used for the sub-basins 10, 15, 17 and 19 is not defined. 20. Figure 2. Sensitivity analysis is missing after SWAT-CUP. 21. Figure 4. Why the upper and lower bound of coef. of PBIAS is only ± 0.15, though the model performance for some sub-basins are more than ± 0.15. 22. Figure 6. How representative is the sampling of only 8 months for turbidity? 23. Figure 12a. Legend for y-axis has typo error. 24. Figures 13, 14 and 17. The legends and axis values are not readable.

---

## Referee Comment (RC2) · Anonymous Referee #2 · 6 Nov 2017

General comments It is an interesting topic that the freshwater quality in different developing scenarios have been simulated using the spatially semi-distributed SWAT mode. But it is more likely one case study in the Cantareira system, Brazil. And an essential improvement should be done further. Specific comments 1. The hypothesis of the research is not clear, and is it "the conversation practices impact hydrological services?"; 2. What is the EbA, and the authors should give the readers more detailed definination. 3. In addition, the paper is so long, and the authors should condense the whole text, as well as the figures and tables. 4. The authors considered the land use scenarios only, but not the climate hydrological factors. 5. The authors should explain the reason why nitrate, TP, and sediments have been select to assess greyWF. 6. Page 11,

[Figure]

Lines295-297, "..., WPL[x,t] exceeds 100%, environmental standards are violated...", it is so subjective. What's your basis? 7. Lines 321, in equation (3), maybe, it is a mistake about the "WPL[x,t]", is it "WPLreference" 8. The authors should separate the results and discussion. Some sentences, for example lines 343-345;349-354;357-360; and so on, should be put into Discussion. The independent discussion could further clearly tell the readers your finding. 9. in Section 3.6, the authors do not depict the results from Figure 17. 10. delete the references from the conclusions. 11. Table 1 should be moved to Supplemental information, or part of Table 1 should be merged in to Table 2. 12. Table 8 should be moved in to Supplemental information. 13. Fig.4, explain the meaning of the lines in the figure. 14. Fig.5, the sentence "Time (horizontal axis) is represented by month/year" is meaningless; further, to provide the meaning of the uncertainty bars and sample numbers. 15. Fig.6, what are the meaning of the "size of circles" and the numbers?

---

## Author Comment (AC1) · 4 Dec 2017

Reviewer Comments 1 (RC1):

"The abstract section could be concise". Answer: We modified abstract. We agree with this comment and corrected the abstract accordingly (see new text).

[General comment], RC1 - "One of the main reasons for the discrepancies between monitoring data/existing literatures and model simulations might be the weakness of SWAT model to capture extreme flows or water yields." Answer: The authors would like to thank the comments from Reviewer #1 and welcome them. We agree with this

general comment. Other detailed responses are described below, as follows.

Original text, Line 55, RC1 – "Colombia (2015, 2014, 2010)" Answer: Corrected in the updated version of the manuscript. Thank you.

Original text, "Lines 58 to 66, RC1 - "Hoekstra et al., 2011" is over cited. Could be rephrased." Answer: This entire paragraph was rephrased. We cited Hoekstra et al. (2001) less times.

Line 141, RC1 –"... run from 2009 to 2014". Answer: Thank you. The new statement is: "The Water Producer/PCJ Project was developed from 2009 to 2014 in the Cantareira System region (Guimarães, 2013), using local actions adopting the concept of Payment for Ecosystem Services-Water [Pagiola et al, 2013; quoted] ".

Line 153, RC1 - "three data collection platforms" their geographic locations could be indicated on the study area map." Answer: The three DCPs are shown in the study area (in Table 1, Table 4 and Figure 8, in the new version of the manuscript).

Line 156, RC1: "the type of secondary data could be clearly indicated." Answer: We appreciate this comment. The explanation to be updated in the new version of the paper appears as follows (because of the extension of these new statements, we suggest including them in a Supplementary Material section, according to the HESS Editor's final decision): "To reduce uncertainty about hydrological scaling effects of EbA through LULC scenarios from 2011 to 2014, we also collected supplementary, secondary data using three strategies. First, we scheduled field observations and interviews with local landowners and farmers who explained their past, present and future (planned) best management practices related to Payment for Ecosystem Services-Water, derived from EbA initiatives from the PCJ-Produtor de Agua Project of the Cantareira System's headwaters [Pagiola et al, 2013, Brazil's Experience with Payments for Environmental Services. Payments for Environmental Services (PES) learning paper; no. 2013-1. World Bank, Washington, DC, World Bank. https://openknowledge.worldbank.org/handle/10986/17854 License: CC BY

3.0 IGO]. This secondary information helped to link LULC derived from EbA/PES-Water with some parameters of selected hydrologic response units (i.e. SWAT-HRUs). These field observations on the local knowledge brought a better understanding about physically-based parameters calibrated regionally, but with unsatisfatory coefficients in some catchments, i.e. Posses Catchment (13-km2 drainage area). Second, we also collected secondary information about the stakeholders' opinions concerning the 23 scenarios we developed in this paper from the multi-agent, multi-level governance of the PCJ-Produtor de Agua Project (municipality, state and national). Due to the states' border between Minas Gerais (MG) and São Paulo (SP), which have different reference standards, the various stakeholders' opinions strongly influenced PES-Water/EbA practices across the transboundary (inter-state) nature of most Cantareira System's catchments. Thus, we undertook extra field visits to evaluate sites with the greatest uncertainty in modelling EbA and LULC scenarios to receive new flow gauging stations. These stations were selected together with representative decision-makers from the states and municipalities that are part of the sub-basins studied (Extrema-MG, Joanópolis-SP, Piracaia-SP and Nazaré Paulista-SP), states (IGAM-MG, SMA-SP and DAEE-SP), federal agencies (ANA-The Brazilian Water Agency, CPRM- Brazilian Geologic Survey, and the National Center for Monitoring & Alerts of Disasters, CEMADEN-MCTIC) and non-government organizations (WWF-Brazil, TNC-Brazil and local initiatives) (see Taffarello et al (2016-b), http://dx.doi.org/10.4236/jep.2016.712152). Third, the aforementioned strategies helped identify, select and prioritize qualitative and quantitative variables to reduce the uncertainties in the generation of pollutant loads under LULC, as proposed by other authors (see e.g. Zaffani et al, 2015; doi:10.4172/2161-0398.1000173, quoted in the references). These secondary data revealed the most viable conditions for nested catchment experiments to monitor and test hypotheses through a scenario-intercomparison modelling of upstream areas of the Jaguari-Jacareí, Cachoeira and Atibainha reservoirs, and are updated regularly by official agencies with open access repositories of hydrological databases, such as ANA (http://hydroweb.ana.gov.br) and CEMADEN

(http://www.cemaden.gov.br/pluviometros-automatico/)".

Lines 252-255, RC1: "Besides adopting from the existing literatures, implementing sensitivity analysis could be recommended in order to select model parameters." Answer: We appreciate this comment. The explanation to be updated in the new version of the paper appears as follows (because of adding these new statements, we suggest including them in a Supplementary Material section, according to the HESS Editor's final decision): "Modelling parameters for water yield calibration were selected not only by consulting the SWAT literature [i.e. Arnold et al, 2012; Bressiani et al, 2015; Fukunaga et al, 2015; Gassman et al, 2007; see more explanations for other review comments below], but also by performing supervised analysis and comparing parameters from recent literature [i.e. Francesconi, W., R. Srinivasan, E. Pérez-Miñana, S.P. Willcock, M. Quintero. 2016]. "Using the Soil and Water Assessment Tool (SWAT) to model ecosystem services: A systematic review", Journal of Hydrology 535 (2016) 625–636. DOI:10.1016/j.jhydrol.2016.01.034, and Monteiro, J. A. F., Kamali, B., Srinivasan, R., Abbaspour, K., and Gücker, B. (2016) "Modelling the effect of riparian vegetation restoration on sediment transport in a human-impacted Brazilian catchment", Ecohydrol., doi: 10.1002/eco.1726, now quoted] and even from consulting the USP open access repository [see studies by Rodrigues, 2014, www.teses.usp.br/teses/disponiveis/18/18138/tde-18122014-094354/pt-br.php; Bressiani, 2016, www.teses.usp.br/teses/disponiveis/18/18138/tde-04042017-155701/pt-br.php, and Mohor, 2016, www.teses.usp.br/teses/disponiveis/18/18138/tde-23032017-102949/pt-br.php]. First, in spite of a much larger list of suggested parameters for modelling goals proposed by Bressiani (2016; quoted), our regional sensitivity analysis followed the recommendations of the theory and practice of mapping ecosystem services using Tier 1 and Tier 2 models [see Mendoza et al, 2012, Ch. 3, in Kareiva et al(eds), 2012; ISBN 978-0-19-958899-2] constrained by the short time series monitored for all sites, with inequal quantitative assessment, seasonality and scale effects. Second, based on the studies carried out by Rodrigues et al [2014, doi:10.1002/2013WR014274, 2015, doi: 10.1002/2014WR016691],

Bressiani et al [2015, doi: 10.3965/j.ijabe.20150803.1765] and Mohor & Mendiondo [2017, doi: 10.1016/j.ecolecon.2017.04.014], we selected 18 SWAT parameters and their initial range of combinations, as follows: Available water capacity, Moist bulk density, Saturated hydraulic conductivity, Baseflow alpha factor, GW-Revap Coefficient, Groundwater delay time, Deep aquifer percolation fraction, Threshold depth of water in the shallow aquifer for "revap" or percolation to the deep aquifer to occur, Soil evaporation compensation factor, Plant uptake compensation factor, Manning's roughness for the main channel, Effective hydraulic conductivity in the main channel, Maximum canopy storage, Manning's for overland flow, Average slope steepness, Initial SCS CN (for antecedent moisture condition 2), and Surface runoff lag coefficient. Thirdly, in-situ field validation tests were developed through experimental campaigns to test the limits of variation of streamflow and water quality (see explantations below).

Lines 276 to 279, RC1 - "It is known that SWAT model is not for extreme flows and hence water quality parameters." Answer: We agree with this comment. For EbA scenario purposes, we planned to set up field investigations and SWAT calibrations [see Figure 5, HESSD paper] using the extreme conditions of 2013–14 drought through quali-quantitative freshwater monitoring at the headwaters of the Cantareira System, quoted in this paper [see i.e. Tafarello et al, 2016; doi: 10.1080/02508060.2016.1188352]. This evidence outlined water quality results from 17 catchments, showing regional behaviour for water quality loads in drainage areas (ranging 0.66–925 km2) for future modelling parameterization through SWAT for EbA scenarios purposes. We experimentally sampled water quality parameters of pH, water temperature, electrical conductivity, turbidity, biological oxygen demand (BOD), chemical oxygen demand (COD), total solids (TS), NO3, NO 2, PO 4, thermotolerant coliforms and Escherichia coli, in several catchments, varying the drainage area, the land use and land cover, which helped us to address the uncertainty and complexity of factors affecting SWAT parameter selection. Moreover, a summary of these results are detailed in Table 4 (this HESSD paper).

Line 299 (RC1) could be moved to line 298. Answer: It was corrected in the updated version of the manuscript. Thank you.

Lines 310 (RC1) could be moved to line 309. Answer: It was corrected in the updated version of the manuscript. Thank you.

Lines 322 (RC1) could be moved to line 321. Answer: It was corrected in the updated version of the manuscript.

Lines 455 to 456 (RC1) should be written with appropriate multiplication sign. Answer: Rewritten as: "... The 52% decrease of water yield between S1 (1990) and S2 (2010) scenarios, as $(14.9 - 31.3)/31.3 \times 100)$ might be related to a marginal increase in the Eucalyptus cover..."

Line 514, RC1: "It would be useful to relate spatially the sub-basins in which the differences in land-use/land-cover are the greatest and the water yield, nitrate, total phosphorus and sediments yield differences are evident. For instance providing maps which indicate temporal changes in LULC and corresponding changes in water quality parameters considered." Answer: We appreciate this comment. The explanation to be updated in the new version of the paper appears as follows (because of adding these new statements, we suggest including them in a Supplementary Material section, according to the HESS Editor's final decision): "Due to the significant variabilities among selected basins where in-situ monitoring was developed for EbA scenario purposes, and because we have not performed field validation in all distributed HRU (hydrologic response units), we decided not to show the regional results through maps. Whatever interpolation technique used, it will not be able to catch the inherent ground-context heteregeneity, and physically-based characteristics of high-variability functionality of these subtropical catchments. Instead, we performed an initial analysis of clustering similar responses from catchments with the most plausible explanations as follows. On the one hand, evidence of SWAT modelled scenarios showed two groups of river basins under EbA scenarios, with distinct land use change of native forest fractions (NF%).

Our results show Group 1, with 11 of the studied basins, with native forest recovery using EbA (S2+EbA), as well as an intermediate land use fraction as follows: NF%(S2)< NF%(S2+EbA)<NF%(S1). In turn, Group 2 of 9 river basins showed a progressive growing fraction of native forests across scenarios, with best EbA land use impacts, as follows: NF%(S1)< NF%(S2)<NF%(S2+EbA). The basins from Group 1 are mainly located close to both urban settlements and Eucaliptus plantations in Northwestern headwaters, where conservation projects have minimum adherence and no significant effect on LULC and, therefore, on SWAT outputs (see Figure 3). Moreover, the catchments from Group 1 are mainly located in Eastern and Southeastern areas (Figure 3), where there are more EbA projects of PCJ-Produtor de Agua. On the other hand, the greatest impacts in water yield are inversely correlated with land-uses and water pollutant quality, but with high non-linear relationships and without explicit regional factors (see Figure 11). For an integrated assessment of hydro-services, it is worth noting that phosphorus, nitrate and sediment yields have spatio-temporal changes of load production across scenarios S1, S2 and S2+EbA, which would be better understood in selected catchments, namely Alto Jaguari and Domithildes.

Line 533, RC1: "Reason for selecting the two sub-basins among the 20 sub-catchments?" Answer: We appreciate this comment. The explanation to be updated in the new version of the paper appears as follows (because of adding these new statements, we suggest including them in a Supplementary Material section, according to the HESS Editor's final decision): "These two catchments were selected regarding the different groups identified in this study, contrasting the outputs from 20 sites: Upper Jaguari is selected from Group 1 and Domithildes is selected from Group 2 [see comment 12]. Moreover, we studied the following variables in the two selected catchments. First, we analysed the fraction of water yield affected by the grey water footprint for nitrate (ca. 0.08 to 3.9 mg/L), total phosphorous (from 0.02 to 1.2 mg/L) and sediments (approx. 0.03 to 250 mg/L). These concentrations represented dilution demands between 0.1 % to close to 1000 % of simulated water yield for a wide range, in between 10 to 500 km2 [see Figure 12, this HESSD paper]. Second, these demands depended

on: the native forest cover [i.e. in Figure 9, with S1 for year 1990, S2 for year 2010 and S2+EbA for year 2035], the flow duration curves under three LULC scenarios at 20 headwaters [Fig. 10], and the scaling effects of EbA actions on drainage areas [ranging from the small Domithildes catchment of 9.9 km2 to the medium-sized Alto Jaguari catchment of 302 km2]. These factors clearly affected (a) the fraction of water yield affected by the GWF-NO3, GWF-TP and GWF-Sed, and (b) the reference flows in duration curves, both in streamflow and in pollutant loads, especially for low-flows (higher duration probabilities [see Fig. 13 and 14]. Moreover, the annual regime of water yield of these two selected catchments revealed local constraints in the size of catchments ranging from 10 to 300 km2. Thus, we pointed out the limits for SWAT modeling when using the EbA assessment and PES-Water projects, by using the grey water footprint, ranging from GWF-NO3 below 0.2 m3/s to GWF-TP up to 20 m3/s. These results converged with the general discussion with blue and green water accounting shown in the studies carried out by Rodrigues et al [2014; A blue/green water-based accounting framework for assessment of water security, Water Resour. Res., 50, 7187–7205, doi:10.1002/2013WR014274], now quoted in the references of this manuscript.

Line 535, RC1: "Any statistical relationship between the changes in LULC classes and grey water footprints. For instance multivariate statistical analysis." Answer: For instance, the evidence we modelled using SWAT concerning GWF and LULC was presented in lines 514 to 534 (first version of the manuscript). These results refer to regionally average values (20 catchments), the same test period (8-yr time series tested) and with the fixed time-step modelled (SWAT monthly-basis). On the one hand, native forest land use fractions (NF%) have ranges of 41±14, 39±15% and 44±16 %, and were related to GWF-NO3 of 0.68±0.6, 0.28±0.1, and 0.44±0.1, for S1(1990), S2(2010) and S2+EbA(2035) scenarios, respectively. On the other hand, medium-sized vegetation land use fraction (native, eucaliptus and orchard) ranged between 46%, 53% and 62% for the same scenarios, respectively, not showing a trend. For GWF-TP and GWF-Sed, the values differ in absolute terms and the averaged ratios of GWF/Water Yield also changed. In spite of the high variability of responses, and

small period of testing, we recommend future field campaigns and further multivariate statistical analysis, but they are out of the scope of the present manuscript.

Lines 544 to 555, RC1: "As one-third of the SWAT simulation are low-flow or drought years. It is known that SWAT model is weak in capturing extreme flows. One of the reasons for the discrepancy between monitoring data and model simulation might not the weakness of the SWAT model to represent low-flows?" Answer: We agree with these comments. On the one hand, recent papers addressing a review of SWAT applications in Brazil outlined the challenges and prospects to reduce the discrepancies between monitoring data and existing (regional) literature and model simulations [i.e. Bressiani et al, 2015; doi: 10.3965/j.ijabe.20150803.1765], quoted in the references. This general review is useful to address model discrepancies in a multilevel approach: quantitative water yield, water quality loads and rainfall-streamflow behaviours at a range of scales during the same period of monitoring and the inherent streamflow variability at these subtropical catchments. Due to this, our strategy selected sites through a nested catchment experiment to study these discrepancies according to the natural hydrological cycle, when possible. On the other hand, we addressed these discrepancies by quantitative calibration with a consecutive freshwater quality calibration. Our evidence showed [see i.e. Fig. 5] that at some drainage areas, between 12 km2 to 508 km2, the SWAT model might underestimate observed streamflows. In three out of four campaigns, the results of both flow rates and nitrate loading (NO3) were very close to the values simulated by SWAT. Only the campaign conducted in May, 2014 demonstrated a significant difference between field validation with SWAT modelling, which may have occurred because of the SWAT limitation in updating loads (water quality parameters) with more prolonged dry periods, as discussed in the papers by Taffarello et al [2016-a; doi: 10.1080/02508060.2016.1188352] and Mohor & Mendiondo (2017; doi: 10.1016/j.ecolecon.2017.04.014, quoted].

Table 1, RC1: "It might be better to replace sub-basin coordinates with key modelling results and/or field observations." Answer: In the new, updated manuscript,

we included new columns, pointing out modelling results and field observations. Table 2, RC1: "Possible reason for model underperformance for some sub-basins?" Answer: As mentioned in the paper, both the Posses catchment and Cachoeira catchment have been constrained by limitations in SWAT modeling set-ups because of: anthropic and ilegal domestic water withdrawals across riversides and margins, with small dams affecting the streamflow regime and in some cases, Eucaliptus sp planted close to river channel during low-flows. Taffarello [2016, quoted] showed this in the open-access repository pictures, which described anthropic impacts on water yield and water withdrawal [see www.teses.usp.br/teses/disponiveis/18/18138/tde-05042017-091421/pt-br.php]. These human-made impacts strongly affected the SWAT underperformance in calibration and validation steps, not only on NASH, NASH-log but also on the PBIAS, especially after long periods of droughts or rainfall anomalies [see Figure 5, this HESSD paper]. Because these human-made interferences come from real situations at the catchments studied, without special SWAT parameterisation and scaling from HRU to the whole catchments, we decided not to reduce both complexity and heteregeneity through a complete, exhaustive sensitivity analysis of SWAT parameters. Instead, we recommend further studies along these lines if new and more field evidence in other catchments is made available. Table 3, RC1: "The selected SWAT parameters are not exhaustive unless sensitivity analysis is conducted." Answer: The main objective of this paper submitted to HESS is not to address sensitivity analysis among SWAT parameters. Instead, we aimed to perform hypothesis tests of scenario intercomparisons, including EbA policies and PES-Water actions, using SWAT pre-calibrated parameters, linked with previous field evidence collected during sampling periods and previous modelling experiences in these basins [i.e. Rodrigues et al, 2014, doi: 10.1002/2013WR014274; Rodrigues et al, 2015; Bressiani et al, 2015, DOI: 10.1002/2014WR016691; Taffarello et al, 2016-a, DOI:10.1080/02508060.2016.1188352; Mohor & Mendiondo, 2017, DOI: 10.1016/j.ecolecon.2017.04.014]. It is worth noting that this paper submitted to HESS is one of the first Brazilian contributions of coupling EbA directives into hydrological

modelling using nested catchment experiments in the Brazilian Atlantic Forest [see Taffarello et al, 2016-b, DOI: 10.1016/j.cliser.2017.10.005], promoting other research groups which might develop furhter modelling hypotheses. Regarding the sensitivity analysis, we proceeded in the calibration process, although it was not exhaustive. On the other hand, and given that SWAT has a very large number of parameters and our experiment involved nested catchments, rather than a single experimental basin, testing all parameters in our study case with EbA would be rather laborious. As mentioned earlier, we consulted previous applications of SWAT in the literature, preferably those in Brazilian basins, to find the most indicated parameters to work on. Based on Fukunaga et al. [2015, DOI: 10.1016/j.catena.2014.10.032], Gassman et al. [2007, DOI: 10.13031/2013.23637], Arnold et al. [2012, DOI: 10.13031/2013.42256] and a good review by Bressiani et al. [2015, quoted], we first selected 18 SWAT parameters with their initial ranges by Rodrigues et al [2014, 2015 quoted]. Then, we made analyses of these 18 parameters in our sub-basins. After analyzing these results, we chose to re-calibrate parameters in some basins. Thus, SWAT-CUP was performed in our tests, with each cycle consisting of 300 runs. In each cycle, we reached new limits for each parameter or stopped tuning a parameter. The number of cycles varied among the sub-basins, from one to five cycles. From all the 20 nested catchments studied, and using the initial 18 SWAT parameters, some sites completed the calibration with 7 calibrated parameters, while others had a total of 17 - out of those initial 18 parameters. From upstream to downstream, after the automatic step, a manual calibration refinement also took place. One example of the range of the final values is shown below in Table A.1 (new).

Table A.1: Range of coefficients adopted for calibration in SWAT-CUP and final values found after manual stage calibration. Parameter* Initial (mín MedianÂź mín Chosen mín Chosen máx MedianÂź máx Initial máx a__CANMX.hru 0 0 0 100 60 100 a__Ch_N2.rte -0.0005 0 0 0.28 0.3 0.3 a__CN2.mgt -15 -12 -8.67 10.31 10 15 a__GW_DELAY.gw -15 -3 -4.161 42.69 30 50 a__GWQMN.gw -550 -300 -415.02 360.00 350 450 r__OV_N.hru -0.5 dismissed - dismissed 1 r__SHALLST.gw

-0.5 -0.3 -0.08 0.39 0.4 0.6 r__SOL_AWC().sol -0.5 -0.25 -0.42 0.29 0.33 0.5 r__SOL_BD(1).sol -0.2 -0.15 -0.19 0.18 0.2 0.4 r__SOL_K().sol -0.4 -0.27 -0.32 0.35 0.37 0.5 v__Alpha_BF.gw 0.01 0.02 0.001 0.049 0.05 0.1 v__Ch_K2.rte 0 0 0 36.74 30 130 v__EPCO.hru 0.4 0.4 0.85 - [2] 1 1 v__ESCO.hru 0.4 0.7 0.69 0.95 0.95 0.95 v__GW_REVAP.gw 0.02 0.02 0.01 0.18 0.2 0.2 v__RCHRG_DP.gw 0.01 0.01 0.05 0.68 0.5 1 v__REVAPMN.gw 0 500 539.28 959.28 1000 1000 v__SURLAG.hru 0.01 1.5 0.97 5.53 4 5 IPET (0) Priestley-Taylor Legends: "1": "median" of the limits adopted in following runs in SWAT-CUP. Manual calibration could overcome these limits; "2": only one sub-basin had EPCO modified. * a_ stands for "added" value, i.e. the final value in each feature (e.g. each HRU) is the original value plus the calibrated coefficient; r_ stands for ratio, i.e. the final value in each feature is the original value times 1+ the calibrated coefficient; v_ stands for value, i.e. the final value of the feature is the calibrated coefficient. References cited: Arnold, J. G.; Moriasi, D. N.; Gassman, P. W.; Abbaspour, K. C.; White, M. J.; Srinivasan, R. et al. (2012): SWAT. Model Use, Calibration, and Validation. Em: Transactions of the ASABE 55 (4), pág. 1491–1508. DOI: 10.13031/2013.42256.

Bressiani D A, Gassman P W, Fernandes J G, Garbossa L H P, Srinivasan R, Bonumá N B, et al. (2015): Review of Soil and Water Assessment Tool (SWAT) applications in Brazil: Challenges and prospects. Em: Int J Agric & Biol Eng, 8(3), pág. 9–35. DOI: 10.3965/j.ijabe.20150803.1765.

Fukunaga, Danilo Costa; Cecílio, Roberto Avelino; Zanetti, Sidney Sára; Oliveira, Laís Thomazini; Caiado, Marco Aurélio Costa (2015): Application of the SWAT hydrologic model to a tropical watershed at Brazil. Em: CATENA 125, pág. 206–213. DOI: 10.1016/j.catena.2014.10.032.

Gassman, P. W.; Reyes, M. R.; Green, C. H.; Arnold, J. G. (2007): The Soil and Water Assessment Tool. Historical Development, Applications, and Future Research Directions. Em: Transactions of the ASABE 50 (4), pág. 1211–1250. DOI: 10.13031/2013.23637.

Rodrigues, D.B.B.; Gupta, H.V.; Mendiondo, E. M. (2015): Assessing uncertainties in surface water security: An empirical multimodel approach. Water Resources Research, 51 (11): 9013–9028. DOI: 10.1002/2014WR016691.

Rodrigues, D.B.B.; Gupta, H.V.; Mendiondo, E. M. (2014): A blue/green water-based accounting framework for assessment of water security. Water Resources Research, 50 (9): 7187–7205. DOI: 10.1002/2013WR014274.

Table 5, RC1: "I would like to see additional column indicating the Hydrologic Services Index. The symbol used for the sub-basins 10, 15, 17 and 19 is not defined." Answer: We answered this comment, including this HSI value as a new column. The symbol used for sub-basins 10, 15, 17 and 19 was a typing error. We appreciate your comment. Thank you.

Figure 2, RC1: "Sensitivity analysis is missing after SWAT-CUP". Answer: In this manuscript, as mentioned, we followed a step-by-step, but not exhaustive, calibration procedure using collection and assessment of data, understanding the watersheds, identifying and selecting sites and periods to calibrate and validate, defining calibration methods, objective functions and evaluation metrics, main water balance components, with volumes and process representations, defining parameters and ranges of variability, sensitivity analysis, calibration, validation, cross validation and uncertainty analysis (Bressiani, 2016, quoted; Mohor, 2016, quoted). As previously mentioned, we also consulted former SWAT modelling strategies used in these basins, available in the open repository by Mohor [2016; www.teses.usp.br/teses/disponiveis/18/18138/tde-23032017-102949/pt-br.php], Bressiani [2016; www.teses.usp.br/teses/disponiveis/18/18138/tde-04042017-155701/pt-br.php] and Rodrigues [2014; http://www.teses.usp.br/teses/disponiveis/18/18138/tde-18122014-094354/pt-br.php]. In our paper, we addressed a calibration stage of SWAT-CUP (Calibration and Uncertainty Programs) software and SUFI-2 (Sequential Uncertainty Fitting) method. SUFI-2 is based on Latin Hypercube sampling [Abbaspour et al, 2015; quoted in the references). After this automatic stage, a finer adjustment using manual calibration was made, following the recommendations of Mohor (2016) and Mohor & Mendiondo (2017; DOI: 10.1016/j.ecolecon.2017.04.014), quoted in the references. For more in-depth sensitivity analysis of SWAT parameters, we recommend Bressiani (2016) who proposed not only a new systematic procedure for calibrating the SWAT model in complex basins, but also a search for a better SWAT performance and reduced optimization time, using different calibration methods on different watershed locations. Moreover, Rodrigues [2014, Table 2.3, page 56] adjusted some parameters for nested catchments in the Cantareira System (CN2, Canmx, OV_N, SOL_K, SOL_AWC), according to land use classes.

Figure 4, RC1: "Why the upper and lower bound of coef. of PBIAS is only ± 0.15, though the model performance for some sub-basins are more than ± 0.15." Answer: We appreciate your comment, thank you. We corrected Figure 04. Figure 6, RC1: "How representative is the sampling of only 8 months for turbidity?" Answer: During the 2013/2014 field campaigns across all the nested catchments presented here, the turbidity ranged between extremes of 1 and 300 NTU, with median values close to 11 NTU. These high variability captured ranges of in-situ monitored instantanous mean cross-section velocities below 1 m/s and specific streamflows ca. 0.001 to 0.025 m3/s/km2. These values captured approximate flow discharges in the range of 5% and 96% of probability of regional flow duration curves, and also affected the variability of the turbidity of water quality. Moreover, these ranges were observed during the 2013/2014 anomalous rainy season, alternating heavy rains and dry periods, in both reference catchments with EbA initiatives and impacted catchments with land-use changes. Due to this, we understand, in spite of having sampled only 8 months of monitoring, observed turbidity is not biased and could represent the conditions for using EbA hypothesis for the scenarios we tested. More details of experimental sampling and observational schemes are explained in Taffarello et al [2016-a; quoted]. Figure 12-a, RC1: "Legend for y-axis has typo error." Answer We appreciate your comment. It was corrected in the updated version Fig. 12-a.

Figures 13, 14 and 17, RC1: "The legends and axis values are not readable." Answer: The legends of Figures 13, 14 and 17 were increased in size and readability is now much better. We appreciate your feedback concerning this correction.

Please also note the supplement to this comment:
https://www.hydrol-earth-syst-sci-discuss.net/hess-2017-474/hess-2017-474-AC1-supplement.pdf

---

## Author Comment (AC2) · 4 Dec 2017

Responses to Reviewer Comments # 2 (RC2):

RC2 – "The hypothesis of the research is not clear, and is it "the conversation practices impact hydrological services?" Answer: The authors would like to thank the comments from Reviewer 2 and welcome them. The hypothesis of the paper is related to how conservation practices addressed by EbA impact hydrology and the ecosystem services, such as maintaining, restoring or improving both the water yield and the freshwater quality, using ecohydrological modeling in different catchment scales. On the other hand, we hypothesized that incentives of EbA policies can affect water yield and wa-

ter quality through non-linear tradeoffs, with high spatiotemporal complexity, which can be assessed by modeling, but previously supported by in-situ monitoring variables for setup boundary conditions of simulation runs. We enhanced these staments in the updated version of the manuscript, refining the statement written previously in lines 87 to 91.

RC2 - What is the EbA, and the authors should give the readers more detailed defi-nition. Answer: The concept of Ecosystem-based Adaptation (EbA) is addressed as 'using biodiversity and ecosystem services to help people adapt to the adverse effects of climate change', which was defined by the Convention on Biological Diversity – 10th Conference of the Parties (CoP) (CBD, 2010, quoted). The payment for ecosystem services is known as a method of EbA. Detailed definitions of EbA applied to the Cantareira's Headwaters (the study area of this manuscript) can be found in our most recent article: Taffarello et al., 2017 [Climate Services (2017), http://dx.doi.org/10.1016/j.cliser.2017.10.005].

RC2 - In addition, the paper is so long, and the authors should condense the whole text, as well as the figures and tables. Answer: Regarding this specific comment, a new, updated version was written, moving some tables and figures to the Supplementary Material. With these actions, the new manuscript has decreased the number of words and graphical elements, but maintained only essential statements and new answers for specific revisions.

RC2 - The authors considered the land use scenarios only, but not the climate hydrological factors. Answer: Due to the high complexity of the interaction and coupling drivers of the climate-soil-water-human nexus, the main objective of the paper aims to only test hypothesess of changes in land use, with adaptation measures PES-Water from EbA options policies. Climate change scenarios will be included in a forthcoming paper. This subject is not relevant to this paper. Some evidence of climate change in hydrological factors, including sensitivity analysis of water withdrawal scenarios, and economic indicators in the Cantareira System's catchments throughout 2000-2100 scenarios can be found in Mohor & Mendiondo (2017; 10.1016/j.ecolecon.2017.04.014, quoted).

RC2 - The authors should explain the reason why nitrate, TP, and sediments have been select to assess greyWF. Answer: SWAT model outputs perform different water quality variables (see Arnold et al, 2005; Bressiani, 2016; quoted). Here we chose to evaluate the greyWF through modeling because these are some freshwater quality variables we had previously sampled in experiments, since such variables are useful for proper SWAT parameterization (see Taffarello et al, 2016-a; quoted). Using a higher number of freshwater variables, however, can make the modelling evidence (on hypothesis testing with EbA) either over-parameterised for analytical purposes or even excessively-detailed for current Brazilian standards of freshwater classification—i.e. with some outputs of freshwater quality variable 1 being above the standards, with variable 2 being below the standards, making it harder for decision-making and planning). Moreover, the high uncertainty in hydrological responses of pollutant loads observed in nested catchment experiments under land change in Brazilian biomes (see Zaffani, A G, Cruz N, Taffarello, D, Mendiondo, E M (2015), Uncertainties in the Generation of Pollutant Loads using Brazilian Nested Catchment Experiments under Change of Land Use & Land Cover. J. Phys Chem Biophys, doi: 2015.10.4172/2161-0398.1000e123; now quoted in the references of the updated version) recommend more parsimonious monitoring and modeling tests to study potential tradeoffs with conservation practices and economic incentives such as EbA.

Page 11, Lines 295-297: RC2 - "WPL[x,t] exceeds 100%, environmental standards are violated...", it is so subjective. What's your basis?. Answer: We appreciate this comment. Following several authors (see Hoekstra et al, 2011; quoted), there is not an upper limit for GWF; it depends on the level of polluted loads being transported into the streamflow. These loads originate from coupling the natural and antropic hydrosedimentological cycles, from the headwaters to the outlet of the basin. Alloctonous and autoctonous loads transported in the main flow, either during floods or even during low-flows, as during the annual flow regime, represent the pollution demand (the numerator of Equation 1, line 298). Otherwise, the dillution capacity of the river flow is represented by the annual flow regime, i.e. related to the mean water yield (the denominator of Equation 1). Due to this, demand can potentially grow beyond the capacity, "violating" the real dillution capacity or autodepuration of a rivercourse. Another water security index relating this river demand-and-capacity can be found in studies carried out by Rodrigues et al [2014; 2015; also quoted]. Because the pollution load thresholds are being monitored not for a unique, isolated quality variable, but for many of them, also with different thresholds of Brazilian standards, Equation 1 needs further development to represent a weighted-threshold, or composite-threshold, to discuss EbA policies through hydrological modeling and scenarios.

Lines 321, RC2- in equation (3), maybe, it is a mistake about the "WPL[x,t]", is it "WPLreference". Answer: We appreciate the reviewer's comment. According to Equation 3, using a regional basis of intercatchment comparison with a proper non-dimensionality, WPL[x,t] represents the composite threshold of any catchment studied, and regionally compared with the reference catchment (WPLcomposite,ref , in relative terms as in percentage). By doing so, Equation 3 can express how the HSI (hydrologic system index), alternatively and regionally, would point out more healthy catchments (HSI < 100%, where EbA outputs through hydrological modeling are more evident), and other catchments where insufficient EbA effects arise. This approach could help decision-making processes concerning Brazilian freshwater standards [see i.e. http://www.mma.gov.br/port/conama/], where multi-parameterization or variables are combined for testing scenarios of land-use and planning. These standards are also compared with state standards and local agencies, such as CETESB [www.cetesb.sp.gov.br] and DAEE [http://www.daee.sp.gov.br/] in São Paulo and IGAM [http://igam.mg.gov.br/] in Minas Gerais; the two neighbor states share these Cantareira System catchments. Furthermore, and because all these agencies use indices for freshwater health, HSI might help to identify regional intercomparisons, both from monitoring and from modelling scenarios, concerning WPLreference and EbA

policies.

RC2 - The authors should separate the results and discussion. Some sentences, for example lines 343-345;349-354;357-360; and so on, should be put into Discussion. The independent discussion could further clearly tell the readers your finding. Answer: We appreciate this comment. We rewrote these lines to help the reader understand our findings, but we were careful not to exceed the maximum number of words in the manuscript.

RC2 - in Section 3.6, the authors do not depict the results from Figure 17. Answer: We appreciate this review. In Section 3.6, we added extra statements about the comparative results of Figure 17 in the new version of the paper as follows [because of the extension of these new statements, we suggest including them in a Supplementary Material section, according to the Editor's final decision). "Figure 17 depicts a summary of monitored and modelled water yield observations and scenarios compared with EbA and GWF outputs in the catchments studied in the Cantareira System. The main bold, vertical, dotted line represents the regional mean water yield, compared with water yields from simulated scenarios, also including their respective GWFs. This figure clearly points out six different conditions, labelled with letters (A, B, C, D, E and F), which configurate potential scenarios of water security according to land-use change and insecurity thresholds, also showing tradeoffs between the water yield and grey water footprint outputs, explained in the text.

RC2 - Delete the references from the conclusions. Answer: Thank you for ths observation. We corrected the conclusions, without citing any references.

RC2 - Table 1 should be moved to Supplemental information, or part of Table 1 should be merged in to Table 2. Answer: We appreciate this comment. We corrected this and accepted these suggestions, merging and relocating the tables.

RC2 - Table 8 should be moved in to Supplemental information. Answer: There is no Table 8. Maybe Table 4(?). We moved it to the Supplementary Material. Thank you.

RC2 - Fig.4, explain the meaning of the lines in the figure. Answer: Dotted lines represent trend lines for some selected basins illustrated here. Our interest in this figure was to question whether there would be both regional trends or scaling in the calibration coefficients, but not found in this first paper. Regional trends of the calibration can show both limits and uncertainty of modelling complex catchments. Due to space, we decided to omit this figure from the updated version. Fig.5, RC2 - he sentence "Time (horizontal axis) is represented by month/year" is meaningless; further, to provide the meaning of the uncertaintybars andsample numbers. Answer: We appreciate this comment. We corrected the quotation. The uncertainty bars represent the minimum and maximum values of measured streamflow and pollutant loads in a cross section of the river during a field campaign of headwater catchments. The high variance in observations of field evidence explains the greater variability of these headwaters in the Cantareira System Fig.6, RC2 – what are the meaning of the "size of circles" and the numbers? Answer: It is only a representation of a 3-D graph, substituting the 3rd axis with the diameter of the circle proportional to the magnitude of the 3rd variable (in this case, the 3rd variable is the turbidity). The number shows the value of turbidity. Figure 6 shows that, although there was a coherent and proportional relation between the observed mean river velocity and observed specific flow, experimental evidence still depicted outliers, from not only reference catchments with EbA/PES-Water options, but also intervention catchments with no EbA/PES-Water options, reflecting an illustrative example of how complex LULC options from EbA could be exaustively sensed into hydrological parameters and simulated scenarios. For these reasons, we adapted our conclusion and recommendations for further studies about new hypothesis testing, according to the aforementioned answers to the reviewers.

Please also note the supplement to this comment:
https://www.hydrol-earth-syst-sci-discuss.net/hess-2017-474/hess-2017-474-AC2-supplement.pdf

---

## Author Comment (AC3) · 24 Jan 2018

On behalf of all co-authors, I am proud of finishing the due revision of the manuscript (https://www.hydrol-earth-syst-sci-discuss.net/hess-2017-474/) with all reviewers' and editors' comments. However, we can only proceed with the submission after receiving the corrected version checked by an English native translator we contracted. Because today is the deadline (Jan. 23th.), we warmly ask you and the HESS Editorial Board for a kind concession of a few days to finish this English style checking. It will be concluded asap. Should you prefer contact us directly, herewith I share mine: Direct Mobile +55 16 997281438 (Denise Taffarello), or our Skype contact: Eduardo.mario.mendiondo

(my Supervisor and the last author of the manuscript). Looking forward to receiving your comprehension.

Best wishes, Dr. Denise Taffarello.

---

## Author Response (AR1)

Sao Carlos, SP, Brazil, 16 March 2018.

**Dr.** Zhenyao Shen**,**
**Dear Editor**

Please find below the responses to the reviewer comments on the manuscript **hess-2017-474**, entitled **"Modelling freshwater quality scenarios with ecosystem-based adaptation in the headwaters of the Cantareira System, Brazil"** and submitted to the Hydrology and Earth System Sciences (HESS) Journal for possible publication.

On behalf of my co-authors, I would like to express my thanks for the reviewers comments and corrections, which have considerably contributed to improve the manuscript. We have included all the modifications requested in the Second Review. These changes and a point-by-point response to the reviewers are described below. Also, we have highlighted the modified passages in the text.

In this new version, we better explained several parts of the text to become it easier to understand. Moreover, the final text was revised by a professional service of English language editing, the Native English Speaker **Jane Godwin Coury**.

We hope that the manuscript – which aims to compare freshwater quality scenarios under different land-use/land-cover (LULC) change, one of them related to best management practices in subtropical headwaters, using the spatially semi-distributed SWAT model in Brazilian subtropical catchments ranging from 7.2 to 1037 km$^2$- can help public-and-private partnerships empowering river basin committees for better decision-making and will be of interest to the HESS journal's broad readership.

Looking forward to a positive reply,

Sincerely,

**Dr. Denise Taffarello**.
Post-doctoral researcher at University of Sao Paulo

 Reviewer Comments 1 (RC1)

"The abstract section could be concise". **Answer: Modified abstract.** We agree with this
commentary and corrected the abstract accordingly (see new text)

**[General comment],** RC1 - "One of the main reasons for the discrepancies between
monitoring data/existing literatures and model simulations might be the weakness of SWAT
model to capture extreme flows or water yields." **Answer:** The authors deeply thank and
welcome the comments of Reviewer #1. We agree with this general comment. Other detailed
responses are described below, as follows.

**Original text, Line 55,** RC1 – "Colombia (2015, 2014, 2010)" **Answer:** Corrected in the
updated version of the manuscript. Thank you.

**Original text, "Lines 58 to 66**, RC1 - "Hoekstra et al., 2011" is over cited. Could be
rephrased." **Answer:** This entire paragraph was rephrased, dropping out the overcitation of
Hoekstra et al (2001)'s work.

**Line 141,** RC1 –"... run from 2009 to 2014". **Answer:** Thank you. The new statement is:
"The Water Producer/PCJ  Project was developed in the period 2009-2014 in the Cantareira
System region (Guimarães, 2013), using EbA scenarios and through local actions through the
concept of Payment for Ecosystem Services-Water [Pagiola et al, 2013; quoted] "

**Line 153,** RC1 - "three data collection platforms "their geographic locations could be
indicated on the study area map." **Answer:** The three DCPs are indicated on the study's area
(in Table 1, Table 4 and Figure 8, new version of the paper).

**Line 156,** RC1: "the type of secondary data could be clearly indicated." **Answer:** We
appreciate this comment. The explanation to be updated in the new version of the paper
appears as follows (because of the extension of these new statements, we suggest worth
appending them in a Supplementary Material section, according to HESS Editor final
decision): "To reduce uncertainty about hydrological scaling effects of EbA through LULC
scenarios, in the period 2011-2014 we also collected supplementary, secondary data through
three strategies. First, we scheduled surveillance and interviews with local owners and
farmers who explained their past, present and future(planned) best management practices
related to Payment for Ecosystem Services-Water, derived from EbA initiatives, of PCJ-
Produtor de Agua Project of Cantareira System´s headwaters [Pagiola et al, 2013, Brazil's
Experience with Payments for Environmental Services. Payments for Environmental Services
(PES) learning paper;no. 2013-1. World Bank, Washington, DC, World Bank.
https://openknowledge.worldbank.org/handle/10986/17854 License: CC BY 3.0 IGO]. These
secondary information helped on linking LULC derived from EbA/PES-Water with some
parameters of selected hydrologic response units (i.e. SWAT-HRUs). These surveillance on
local knowledge brought a better understanding on physically-based parameters calibrated
regionally, but with unsatisfatory coefficients in some catchments, i.e. Posses Catchment (13-
$km^2$ drainage area). Second, we also gathered secondary information about the scenarios'
vision storylines from the multi-agent, multi-level governance of PCJ-Produtor de Agua

Project (municipality, state and national). Because of the states' border in between Minas
Gerais (MG) and São Paulo (SP) with different reference reference standards, these multi-
agent vision have strongly incluenced PES-Water/EbA practices across the transboundary
(inter-state) nature of most Cantareira System's catchments. Thus, we performed extra field
visits to select sites, with higher uncertainty in modelling EbA and LULC scenarios, to
receive new flow gauging stations selected in companion with decision-makers representative
of neighbor municipalities (Extrema-MG, Joanópolis-SP, Piracaia-SP and Nazaré Paulista-
SP), states (IGAM-MG, SMA-SP and DAEE-SP), federal agencies (ANA-The Brazilian
Water Agency, CPRM- Brazilian Geologic Survey, and the National Center for Monitoring &
Alerts of Disasters, CEMADEN-MCTIC) and non-government organizations (WWF-Brazil,
TNC-Brazil and local initiatives) (see Taffarello et al (2016-b),
http://dx.doi.org/10.4236/jep.2016.712152). Third, the fore-mentioned strategies aided on the
identification, selection and priorization of qualitative and quantitative variables to reduce the
uncertainties in the generation of pollutant loads under LULC, as proposed by other authors
(see i.e. Zaffani et al, 2015; doi:10.4172/2161-0398.1000173, quoted in the references). These
secondary data revealed most viable conditions for nested catchment experiments to monitor
experiments and test hypotheses through a scenario-intercomparion modelling of upstream
areas of the Jaguari-Jacareí, Cachoeira and Atibainha reservoirs, being updated regularly by
official agencies with open access reposior of hydrological database, like ANA
(http://hydroweb.ana.gov.br) and CEMADEN (http://www.cemaden.gov.br/pluviometros-
automatico/)".

**Lines 252-255**, RC1: "Besides adopting from the existing literatures, implementing
sensitivity analysis could be recommended in order to select model parameters." **Answer:** We
appreciate this comment. The explanation to be updated in the new version of the paper
appears as follows (because of the extension of these new statements, we suggest worth
appending them in a Supplementary Material section, according to HESS Editor final
decision): "The selection of modelling parameters for water yield calibration was developed
not only through consulting on SWAT literature [i.e. Arnold et al, 2012; Bressiani et al, 2015;
Fukunaga et al, 2015; Gassman et al, 2007; see more explanations for other review comments
below] but also performing supervised analysis and comparison of parameters, from recente
literature [i.e. Francesconi, W., R. Srinivasan, E. Pérez-Miñana, S.P. Willcock, M. Quintero.
2016. Using the Soil and Water Assessment Tool (SWAT) to model ecosystem services: A
systematic review. *Journal of Hydrology* 535 (2016) 625–636.
DOI:10.1016/j.jhydrol.2016.01.034, and Monteiro, J. A. F., Kamali, B., Srinivasan, R.,
Abbaspour, K., and Gücker, B. (2016) Modelling the effect of riparian vegetation restoration
on sediment transport in a human-impacted Brazilian catchment. Ecohydrol., doi:
10.1002/eco.1726, now quoted]] and even from consultation of USP open access repository
[see i.e. works of Rodrigues, 2014, www.teses.usp.br/teses/disponiveis/18/18138/tde-
18122014-094354/pt-br.php; Bressiani, 2016,
www.teses.usp.br/teses/disponiveis/18/18138/tde-04042017-155701/pt-br.php, and Mohor,
2016, www.teses.usp.br/teses/disponiveis/18/18138/tde-23032017-102949/pt-br.php]. Firstly,
in spite of a much larger list of suggested parameters for modelling goals proposed by

Bressiani (2016; quoted), our regional sensitivity analysis followed the recommendations of theory and practice of mapping ecosystem services using Tier 1 and Tier 2 models [see Mendoza et al, 2012, Ch. 3, in Kareiva et al(eds), 2012; ISBN 978-0-19-958899-2] constrained by the short time series monitored for all sites, with inequal quantitative assessment, seasonality and scale effects. Secondly, from the works of Rodrigues et al [2014, doi:10.1002/2013WR014274, 2015, doi: 10.1002/2014WR016691], Bressiani et al [2015, doi: 10.3965/j.ijabe.20150803.1765] and Mohor & Mendiondo [2017, doi: 10.1016/j.ecolecon.2017.04.014] we selected 18 SWAT parameters and their initial range of combinations, as follows: Available water capacity, Moist bulk density, Saturated hydraulic conductivity, Baseflow alpha fator, Threshold depth of shallow aquifer to occur return flow, GW-Revap Coefficient, Groundwater delay time, Deep aquifer percolation fraction, Threshold depth of shallow aquifer to revap or percolation to the deep aquifer, Soil evaporation compensation fator, Plant uptake compensation fator, Manning roughness for the main channel, Effective hydraulic conductivity in main channel, Maximum canopy storage, Manning for overland flow, Average slope steepness, Inicial SCS CN (for antecedente moisture condition 2), and Surface runoff lag coeff. Thirdly, in-situ field validation tests were developed through experimental campaigns to test the limits of variation of streamflow and water quality (see explantations below)

**Lines 276 to 279,** RC1 - "It is known that SWAT model is not for extreme flows and hence water quality parameters." **Answer**: We agree with this comment. For EbA scenarios purposes, we planned set up field investigations and SWAT calibrations [see Figure 5, this HESSD paper] using the extreme conditions of 2013–14 drought through quali-quantitative freshwater monitoring at the headwaters of the Cantareira System, quoted in this paper [see i.e. Taffarello et al, 2016; doi: 10.1080/02508060.2016.1188352]. Those evidences outlined water quality results from 17 catchments, showing regional behaviour for water quality loads in drainage areas (ranging 0.66–925 $km^2$) for future modelling parameterization through SWAT for EbA scenarios purposes. We experimentally sampled water quality parameters of pH, water temperature, electrical conductivity, turbidity, biological oxygen demand (BOD), chemical oxygen demand (COD), total solids (TS), NO3, NO 2, PO 4, thermotolerant coliforms and Escherichia coli, in several catchments, varying the drainage area, the land use and land cover, helped us to face about uncertainty and complexity of factors affecting SWAT parameter selection. Also, a summary of these results are detailed in Table 4 (this HESSD paper).

**Line 299** (RC1) could be moved to line 298. **Answer**: It was corrected in the updated version of the manuscript. Thank you.

**Lines 310** (RC1) could be moved to line 309. **Answer**: It was corrected in the updated version of the manuscript.

**Lines 322** (RC1) could be moved to line 321 **Answer**:It was corrected in the updated version of the manuscript.

**Lines 455 to 456** (RC1) should be written with appropriate multiplication sign **Answer**: Rewritten as: "... The 52% decrease of water yield between S1 (1990) and S2 (2010) scenarios, as (14.9 -31.3)/31.3×100) might be related to a marginal increase of Eucalyptus cover..."

**Line 514**, RC1: "It would be useful to relate spatially the sub-basins in which the differences in land-use/land-cover are the greatest and the water yield, nitrate, total phosphorus and sediments yield differences are evident. For instance providing maps which indicate temporal changes in LULC and corresponding changes in water quality parameters considered."
**Answer**: We appreciate this comment. The explanation to be updated in the new version of the paper appears as follows (because of the extension of these new statements, we suggest worth appending them in a Supplementary Material section, according to HESS Editor final decision): "Because of the significant variabilities among selected basins where in-situ monitoring were developed for EbA/PES' scenarios purposes, and because we have not performed field validation in all distributed HRU (hydrologic response units), we decided not showing regional results through maps. Whichever interpolation techniques would not be able of catch the inherent ground-context heteregeneity, and physically-based characteristics, of high-variability functionality of these subtropical catchments. Instead, we do perform initial analysis of clustering similar responses from catchments with most plausible explanations as follows. On the one hand, evidences of SWAT modelled scenarios showed two groups of river basins under EbA scenarios, with distinct land use change of native forest fractions(NF%). Our results show Group 1, with 11 of studied basins, with native forest recovery using EbA (S2+EbA), with an intermediate land use fraction as follows: NF%(S2)< NF%(S2+EbA)<NF%(S1). In turn, Group 2, of 9 river basins, showed a progressive growing fraction of native forests across scenarios, with best EbA land use impacts, as follows: NF%(S1)< NF%(S2)<NF%(S2+EbA). Basins of Group 1, are mainly located close to both urban settlements and Eucalipto plantation in Northwestern headwaters, where conservation projects have small adherence of landowners to EbA/PES-Water actions in LULC and, doing so, in SWAT outputs (see Figure 3). Moreover, catchments of Group 1 are mainly located in Eastern and Southeastern areas (Figure 3), where EbA projects of *PCJ-Produtor de Agua* are more expressive. On the other hand, the greatest impacts in water yield are inversely correlated with land-uses and water pollutant quality, but with high non-linear relationships and without explicit regional factors (see Figure 11). For an integrated assessment of hydro-services, it is worth noting phosphorus, nitrate and sediment yields have spatio-temporal changes of load production across scenarios S1, S2 and S2+EbA, which would be better understood in selected catchments, namely Alto Jaguari and Domithildes".

**Line 533**, RC1: "Reason for selecting the two sub-basins among the 20 sub-catchments?"
**Answer**: We appreciate this comment. The explanation to be updated in the new version of the paper appears as follows (because of the extension of these new statements, we suggest worth appending them in a Supplementary Material section, according to HESS Editor final decision): "These two catchments were selected regarding the distinct groups identified in this study, contrasting the outputs from 20 sites: Upper Jaguari is selected from Group 1 and

Domithildes is selected from Group 2 [see comment 12]. Moreover, we studied the following
variables in the two selected cachment. First, we analysed the fraction of water yield
compromised by the grey water footprint for nitrate (ca. 0.08 to 3.9 mg/L), total phosphorous
(from 0.02 to 1.2 mg/L) and sediments ( approx. 0.03 to 250 mg/L). These concentrations
represented dilution demands in between 0.1 % to close 1000 % of simulated water yield for a
wide range, in between 10 to 500 km$^2$ [see Figure 12, this HESSD paper]. Second, these
demands depended on: the native forest cover [i.e. in Figure 9, with S1 for year 1990, S2 for
year 2010 and S2+EbA for year 2035], the flow duration curves under three LULC scenarios
at 20 headwaters [Fig. 10], and the scaling effects of EbA actions on drainage areas [ranging
from small catchment of 9.9 km$^2$ of Domithildes to medium-size catchment of 302 km$^2$ of
Alto Jaguari]. These factors clearly affected (a) the fraction of water yield compromised by
the GWF-NO$_3$, GWF-TP and GWF-Sed, and (b) the reference flows in duration curves, both
in streamflow and in pollutant loads, especially for low-flows (higher duration probabilities
[see Fig. 13 and 14]. As well, the annual regime of water yield of these two selected
catchments revealed local constraints in the size of catchments ranging from 10 to 300 km$^2$.
Thus, we pointed what limits for SWAT modeling when using the EbA assessment and PES-
Water projects, by using grey water footprint, ranging from GWF-NO3 below 0.2 m$^3$/s to
GWF-TP up to 20 m$^3$/s. These results did converge to the general discussion with blue and
green water accounting showed in former studies of Rodrigues et al [2014; A blue/green
water-based accounting framework for assessment of water security, Water Resour. Res., 50,
7187–7205, doi:10.1002/2013WR014274], now quoted in the references of this manuscript"

**Line 535**, RC1: "Any statistical relationship between the changes in LULC classes and grey
water footprints. For instance multivariate statistical analysis.". **Answer**: For instance, the
evidences we modelled with SWAT about GWF and LULC were presented in between lines
514 and 534 (first version manuscript). These results are regarded to average values
regionally (20 catchments), the same test period (8-yr time series tested) and with fixed time-
step modelled (SWAT monthly-basis). On the one hand, native forest land use
fractions(NF%) have ranges of 41±14, 39±15% and 44±16 %, and were related to GWF-
NO3 of 0.68±0.6, 0.28±0.1, and 0.44±0.1, for S1(1990), S2(2010) and S2+EbA(2035)
scenarios, respectively. On the other hand, high-stand vegetation land use fraction (native,
eucalipto and orchad) ranged in between 46%, 53% and 62%, for the same scenarios,
respectively, not showing a trend. For GWF-TP and GWF-Sed values differ in absolute terms,
and the averaged ratios of GWF/Water Yield also changed. In spite of the high variability of
responses, and small period of testing, we recommend future field campaigns and further
multivariate statistical analysis, but they are out of the scope of the present manuscript.

**Lines 544 to 555**, RC1: "As one-third of the SWAT simulation are low-flow or drought
years. It is known that SWAT model is weak in capturing extreme flows. One of the reasons
for the discrepancy between monitoring data and model simulation might not the weakness of
the SWAT model to represent low-flows?". **Answer:** We agree with these comments. On the
one hand, recent papers addressing a review of SWAT applications in Brazil outlined the
challenges and prospects for reducing the discrepancies between monitoring data and existing
(regional) literatures and model simulations [i.e. Bressiani et al, 2015; doi:

[10.3965/j.ijabe.20150803.1765], quoted in the references. These general review is useful to
address model discrepancies in a multilevel approach: quantitative water yield, water quality
loads and rainfall-streamflow behaviours at a range of scales during the same period of
monitoring and the inherent streamflow variability at these subtropical catchments. For that
reason, our strategy selected sites through a nested catchment experiment to study these
discrepancies according to the natural hydrological cycle, when possible. On the other hand,
we addressed those discrepancies by quantitative calibration with a consecutive freshwater
quality calibration. Our evidences showed [see i.e. Fig. 5] that at some drainage areas, in
between 12 km$^2$ to 508 km$^2$, SWAT model might underestimate observed streamflows. Even
in three of four campaigns, both streamflow quantitative validation and quality (NO$_3$)
simulaton did perform close to SWAT model runs. Only the May,2014 campaign denoted a
higher departure between field validation with SWAT modelling, probably because of SWAT
limitation of updating water quality parameters with the extension of duration time of drought
period as pointed in quoted papers of Taffarello et al [2016-a; doi:
10.1080/02508060.2016.1188352 ] and Mohor & Mendiondo (2017; doi:
10.1016/j.ecolecon.2017.04.014, quoted].

**Table 1**, RC1: "It might be better to replace sub-basin coordinates with key modelling results
and/or field observations."**Answer**: In the new, updated manuscript, we included new
columns, pointing modelling results and field observations.

**Table 2,** RC1: "Possible reason for model underperformance for some sub-basins?" **Answer:**
As mentioned in the paper, both Posses catchment and Cachoeira catchment have been
constrained by limitations in SWAT modeling set-ups because of: anthropic and ilegal
domestic water withdrawals across riversides and margins, with small dams affecting the
streamflow regime and with, some cases, Eucalipto sp planted close to river channel during
low-flows. Taffarello [2016, quoted] showed in the open-access repository pictures which
described    antropic    impacts    on    water    yield    and    water    withdrawal    [see
www.teses.usp.br/teses/disponiveis/18/18138/tde-05042017-091421/pt-br.php].    Those
human-made impacts strongly affected the SWAT underperformance in calibration and
validation steps, not only on NASH, NASH-log but also on the PBIAS, especially after long
period of droughts or rainfall anomalies [see Figure 5, this HESSD paper]. Because these
human-made interference come from real situations at catchments studied, without special
SWAT parameterisation and scaling from HRU to the whole catchments, we decided not
reducing both complexity and heteregeneity through a complete, exhaustive sensitivity
analysis of SWAT parameters.  Instead, we recommend further works in this direction if new,
more field evidences in other catchments would be available.

**Table 3**, RC1: "The selected SWAT parameters are not exhaustive unless sensitivity analysis
is conducted." **Answer:** As explained before, the main objective of this paper submitted to
HESS is not addressing sensitivity analysis among SWAT parameters. Instead, we aimed to
perform hypotheses' tests of scenarios intercomparison, including EbA policies and PES-
Water actions, aided by SWAT pre-calibrated parameters, linked with previous field evidences collected during sampling periods and previous modelling experiences in these basins [i.e. Rodrigues et al, 2014, doi: 10.1002/2013WR014274; Rodrigues et al, 2015; Bressiani et al, 2015, DOI: 10.1002/2014WR016691; Taffarello et al, 2016-a, DOI:10.1080/02508060.2016.1188352; Mohor & Mendiondo, 2017, DOI: 10.1016/j.ecolecon.2017.04.014]. It is worth noting this paper submitted to HESS is one of the first Brazilian contributions of coupling EbA directives into hydrological modelling using nested catchment experiments and monitoring in Brazilian Atlantic Forest [see i.e. Taffarello et al, 2016-b, DOI: 10.1016/j.cliser.2017.10.005, quoted], promoting other research groups which might develop furhter modelling hypotheses. Regarding the sensitivity analysis, as questioned, we proceeded in the calibration process, although not exhaustive. On the other hand, and given that SWAT has a very large number of parameters and our experiment involved nested catchments, rather than a single experimental basin, testing all parameters in our study case with EbA would be rather laborious. As mentioned earlier, we have then consulted previous applications of SWAT in the literature, preferably those in Brazilian basins, to find most indicated parameters to work on. From Fukunaga et al. [2015, DOI: 10.1016/j.catena.2014.10.032], Gassman et al. [2007, DOI: 10.13031/2013.23637], Arnold et al. [2012, DOI: 10.13031/2013.42256], and the good review from Bressiani et al. [2015, quoted], we firstly selected 18 SWAT parameters and with their initial ranges by Rodrigues et al [2014, 2015 quoted]. Then, we ran analysis of these 18 parameters in our sub-basins. After analyzing these preview results, we have chosen to re-calibrate some parameters in some basins. Thus, SWAT-CUP was performed in our tests, with each cycle consisted of 300 runs. In each cycle we reached new limits for each parameter or even stopped tuning a parameter. The number of cycles varied among sub-basins, from one to 5 cycles. From the all 20 nested catchments here studied in Cantareira System and using initial 18 SWAT parameters, some sites ended up the calibration with 7 parameters calibrated, while others had a total of 17 - out of those initial 18 parameters. From upstream to downstream, after the automatic step, a manual calibration refinement also took place. One example of the range of the final values is shown below in Table A.1(new).

Table A.1: Range of coefficients adopted for calibration in SWAT-CUP and final values found after manual stage calibration

| Parameter* | Initial (mín | Median[1] mín | Chosen mín | Chosen máx | Median[1] máx | Initial máx |
|---|---|---|---|---|---|---|
| a__CANMX.hru | 0 | 0 | 0 | 100 | 60 | 100 |
| a__Ch_N2.rte | -0.0005 | 0 | 0 | 0.28 | 0.3 | 0.3 |
| a__CN2.mgt | -15 | -12 | -8.67 | 10.31 | 10 | 15 |
| a__GW_DELAY.gw | -15 | -3 | -4.161 | 42.69 | 30 | 50 |
| a__GWQMN.gw | -550 | -300 | -415.02 | 360.00 | 350 | 450 |
| r__OV_N.hru | -0.5 | dismissed | - | | dismissed | 1 |
| r__SHALLST.gw | -0.5 | -0.3 | -0.08 | 0.39 | 0.4 | 0.6 |
| r__SOL_AWC().sol | -0.5 | -0.25 | -0.42 | 0.29 | 0.33 | 0.5 |
| r__SOL_BD(1).sol | -0.2 | -0.15 | -0.19 | 0.18 | 0.2 | 0.4 |
| r__SOL_K().sol | -0.4 | -0.27 | -0.32 | 0.35 | 0.37 | 0.5 |
| v__Alpha_BF.gw | 0.01 | 0.02 | 0.001 | 0.049 | 0.05 | 0.1 |
| v__Ch_K2.rte | 0 | 0 | 0 | 36.74 | 30 | 130 |
| v__EPCO.hru | 0.4 | 0.4 | 0.85 | - [2] | 1 | 1 |
| v__ESCO.hru | 0.4 | 0.7 | 0.69 | 0.95 | 0.95 | 0.95 |
| v__GW_REVAP.gw | 0.02 | 0.02 | 0.01 | 0.18 | 0.2 | 0.2 |
| v__RCHRG_DP.gw | 0.01 | 0.01 | 0.05 | 0.68 | 0.5 | 1 |
| v__REVAPMN.gw | 0 | 500 | 539.28 | 959.28 | 1000 | 1000 |
| v__SURLAG.hru | 0.01 | 1.5 | 0.97 | 5.53 | 4 | 5 |
| IPET | | | (0) Priestley-Taylor | | | |

Legends: "1": "median" of the limits adopted in following runs in SWAT-CUP. Manual calibration could overcome these limits; "2": only one sub-basin had EPCO modified. * a_ stands for "added" value, i.e. the final value in each feature (e.g. each HRU) is the original value plus the calibrated coefficient; r_ stands for ratio, i.e. the final value in each feature is the original value times 1+ the calibrated coefficient; v_ stands for value, i.e. the final value of the feature is the calibrated coefficient.

References cited:

Arnold, J. G.; Moriasi, D. N.; Gassman, P. W.; Abbaspour, K. C.; White, M. J.; Srinivasan, R. et al. (2012): SWAT. Model Use, Calibration, and Validation. Em: *Transactions of the ASABE* 55 (4), pág. 1491–1508. DOI: 10.13031/2013.42256.

Bressiani D A, Gassman P W, Fernandes J G, Garbossa L H P, Srinivasan R, Bonumá N B, et al. (2015): Review of Soil and Water Assessment Tool (SWAT) applications in Brazil: Challenges and prospects. Em: *Int J Agric & Biol Eng,* 8(3), pág. 9–35. DOI: 10.3965/j.ijabe.20150803.1765.

Fukunaga, Danilo Costa; Cecílio, Roberto Avelino; Zanetti, Sidney Sára; Oliveira, Laís Thomazini; Caiado, Marco Aurélio Costa (2015): Application of the SWAT hydrologic model to a tropical watershed at Brazil. Em: *CATENA* 125, pág. 206–213. DOI: 10.1016/j.catena.2014.10.032.

Gassman, P. W.; Reyes, M. R.; Green, C. H.; Arnold, J. G. (2007): The Soil and Water Assessment Tool. Historical Development, Applications, and Future Research Directions. Em: *Transactions of the ASABE* 50 (4), pág. 1211–1250. DOI: 10.13031/2013.23637.

**Table 5**, RC1: "I would like to see additional column indicating the Hydrologic Services Index. The symbol used for the sub-basins 10, 15, 17 and 19 is not defined." **Answer**: We attended this comment, appending this HSI value as a new column. The symbol used for sub-basins 10, 15, 17 and 19 was a digiting error. We appreciate your review, thank you.

**Figure 2**, RC1: "Sensitivity analysis is missing after SWAT-CUP" **Answer**: In this manuscript, as mentioned, we followed a step-by-step, but not exhaustive, calibration procedure using collection and assessment of data, with understanding of watersheds, identification and selection of sites and periods to calibrate and validate, definition of calibration methods, objective functions and evaluation metrics, main water balance components, with volumes and processess' representations, definition of parameters and ranges of variability, sensitivity analysis, calibration, validation, cross validation and uncertainty analysis (Bressiani, 2016, quoted; Mohor, 2016, quoted). As previously mentioned, we also consulted former SWAT modelling strategies used in these basins, available in open repository of Mohor [2016; www.teses.usp.br/teses/disponiveis/18/18138/tde-23032017-102949/pt-br.php], Bressiani [2016; www.teses.usp.br/teses/disponiveis/18/18138/tde-04042017-155701/pt-br.php] and Rodrigues [2014; http://www.teses.usp.br/teses/disponiveis/18/18138/tde-18122014-094354/pt-br.php]. In our paper, we addressed an stage of calibration of SWAT-CUP (Calibration and Uncertainty Programs) software and SUFI-2 (Sequential Uncertainty Fitting) method. SUFI-2 is based in Latin Hypercube sampling [Abbaspour et al, 2015; quoted in the references). After this automatic stage, a finer adjustement with manual calibration was accomplished, following the recommendations of Mohor (2016) and Mohor & Mendiondo (2017; DOI: 10.1016/j.ecolecon.2017.04.014), quoted in the references. For deeper sensitivity analysis of SWAT parameters we recommend Bressiani (2016) who proposed not only a new systematic procedure for calibrating SWAT model in complex basins, but also a searching for a better SWAT performance and reduced optimization time, using different calibration methods on different watershed locations. Also, Rodrigues [2014, Table 2.3, page 56] adjusted some parameters for nested catchments in Cantareira System (CN2, Canmx, OV_N, SOL_K, SOL_AWC), according to land use classes.

**Figure 4,** RC1: "Why the upper and lower bound of coef. of PBIAS is only ± 0.15, though the model performance for some sub-basins are more than ± 0.15.". **Answer:** We appreciate your comment. We corrected this figure.

**Figure 6,** RC1: "How representative is the sampling of only 8 months for turbidity?" **Answer:** During the 2013/2014 field campaigns across all the nested catchments here presented, turbidity ranged between extremes of 1 and 300 NTU, with median value close to 11 NTU. These high variability captured ranges of in-situ monitored instantanous mean cross-section velocities below 1 m/s and specific streamflows ca. 0.001 to 0.025 $m^3/s/km^2$. These values captured approximately flow discharges in the range of 5% and 96% of probability of regional flow duration curves, and also affected the variability of the turbidity of water quality. Moreover, these ranges were observed during the 2013/2014 anomalous rainy season, with alternance of heavy rains and dry periods, in both reference catchments with EbA initiatives and impacted catchments with land-use changes. For those reasons, we understand, spite of having sampled only 8 months of monitoring, observed turbidity is not biased and could represent the conditions for using EbA hypothesis for the scenarios we tested. More details of experimental sampling and observational schemes are explained in Taffarello et al [2016-a; quoted]

**Figure 12-a** , RC1: "Legend for y-axis has typo error." **Answer:** We appreciate your comment. It was corrected, in the updated version of Figure 12-a

**Figures 13, 14 and 17,** RC1: "The legends and axis values are not readable." **Answer:** The legends of Figures 13, 14 and 17 were augmented in size and readability. We appreciated your feedback for its correction.

**Reviewer Comments # 2 (RC2)**

RC2 – "The hypothesis of the research is not clear, and is it "the conversation practices impact hydrological services?" **Answer**: The authors deeply thank and welcome the comments of Reviewer 2. The working hypothesis of the paper is related to, on the one hand, how conservation practices addressed by EbA impact hydrology and the ecosystem services, like maintaining, restoring or improving both the water yield and the freshwater quality, using hydrological modeling in different catchment scales. On the other hand, we hypothesized incentives of EbA policies can affect water yield and water quality through non-linear tradeoffs, with high spatiotemporal complexity, capable of being assessed by modeling, but previously supported by in-situ monitoring variables for setup boundary conditions of simulation runs. We enhanced these staments in the updated version of the manuscript, refining the statement written previously in between lines 87 to 91.

RC2 - What is the EbA, and the authors should give the readers more detailed definition. **Anwer:** The concept of Ecosystem-based Adaptation (EbA) is addressed as 'using biodiversity and ecosystem services to help people adapt to the adverse effects of climate change' was defined by the Convention on Biological Diversity – 10th Conference of the Parties (CoP) (CBD, 2010, quoted). Detailed definitions of EbA applied to the Cantareira's Headwaters (this paper) can be found in Taffarello et al [2017, Climate Services (2017), http://dx.doi.org/10.1016/j.cliser.2017.10.005]

RC2 - In addition, the paper is so long, and the authors should condense the whole text, as well as the figures and tables. **Answer:** Attending this specific comment, a new, updated version was prepared, translating some tables and figures to Supplement Material. With these actions, the new manuscript has decreased the number of words and graphical elements, but maintained only essential statements and new answers for specific revisions.

RC2 - The authors considered the land use scenarios only, but not the climate hydrological factors. **Answer:** Because the high complexity of the interaction and coupling drivers of the climate-soil-water-human nexus, the main goal of the paper aims to only test hypothesis of changes in land uses, with adaptation measures PES-Water from EbA options policies. Climate change scenarios are being incorporated in a sequential paper, but is out the scope of this presente manuscript. Some evidences of climate change onto hydrological factors, including sensitivity analysis of water withdrawal scenarios, and economic indicators in Cantareira System's catchments throughout 2000-2100 scenarios can be found in Mohor & Mendiondo (2017; 10.1016/j.ecolecon.2017.04.014, quoted)

RC2 - The authors should explain the reason why nitrate, TP, and sediments have been select to assess greyWF. **Answer:** SWAT model outputs perform different water quality variables (see Arnold et al, 2005; Bressiani, 2016; quoted). Here we selected for greyWF through modeling some freshwater quality variables we have previously sampled in experiments, and being usable for a proper SWAT parameterization (see Taffarello et al, 2016-a; quoted). By using a higher number of freshwater variables, however, might the modelling evidences (on
hypothesis testing with EbA) be either over-parameterised for analytical purposes, or even
excessively-detailed for the running Brazilian standards of freshwater classification—i.e. with
some outputs of freshwater quality variable 1 being above the standards, with variable 2 being
below the standards, making it harder for decision-making and planning). Also, the high
uncertainty in hydrological responses of pollutant loads observed in nested catchment
experiments under land change in Brazilian biomes (see Zaffani, A G, Cruz N, Taffarello, D,
Mendiondo, E M (2015) Uncertainties in the Generation of Pollutant Loads using Brazilian
Nested Catchment Experiments under Change of Land Use & Land Cover. *J. Phys Chem*
*Biophys*, doi: 2015.10.4172/2161-0398.1000e123; now quoted in the references of updated
version) recommend more parcimonious monitoring and modeling tests to study potential
tradeoffs with conservation practices and economic incentives like EbA

**Page 11, Lines 295-297:** RC2 - "WPL[x,t] exceeds 100%, environmental standards are
violated...", it is so subjective. What's your basis?. **Answer:** We appreciate this comment.
Following several authors (see Hoekstra et al, 2011; quoted), there is not an upper limit for
GWF; it depends on the level of polluted loads being transported in the streamflow. These
loads are originated from coupling the natural and antropic hydrosedimentological cycles,
from the headwaters to the outlet of the basin. Alloctonous and autoctonous loads transported
in the main flow, either during floods or even during low-flows, as during the annual flow
regime, represent the pollution demand (the numerator of the equation 1, line 298). Otherwise,
the dillution capacity of the river flow is represented by the annual flow regime, i.e. related to
the mean water yield (the denominator of the equation 1). For that reason, demand can
potentially grow beyond the capacity, "violating" the real dillution capacity or autodepuration
of a rivercourse. Other water security index relating these river demand-and-capacity can be
read in the works of Rodrigues et al [2014; 2015; also quoted]. Because the pollution load
thresholds are being monitored not for an unique, isolated quality variable, but for many of
them, also with different thresholds of Brazilian standards, equation 1 needs a further
development to represent a weighted-threshold, or composite-threshold, to discuss EbA
policies through hydrological modeling and scenarios.

**Lines 321,** RC2- in equation (3), maybe, it is a mistake about the "WPL[x,t]", is it
"WPLreference". **Answer:** We appreciate the reviewer's comment. According to the
equation 3, using a regional basis of intercatchment comparion with a proper non-
dimensionality, WPL[x,t] represents the composite threshold of a whichever catchment
studied, and regionally compared with the reference catchment (WPL$_{composite,ref}$ , in
relative terms as in percentage). Doing so, equation 3 can express how HSI (hydrologic
system index), alternatively and regionally, would point more healthy catchments  (HSI <
100%, where where EbA outputs through hydrological modeling are more evident ), and other
catchments where insufficient EbA effects arise. This approach could help decision-making
process of Brazilian freshwater standards [see i.e. http://www.mma.gov.br/port/conama/],
where multi-parameterization or variables are combined for testing scenarios of land-uses and
planning. These standards are also compared with state standards and local agencies, like

CETESB [ www.cetesb.sp.gov.br ] and DAEE [ http://www.daee.sp.gov.br/ ] in São Paulo and IGAM [ http://igam.mg.gov.br/] in Minas Gerais, the two neighbor states sharing these Cantareira System's catchments. Furthermore, and because all these agencies use indices for freshwater health, HSI might help on identifying regional intercomparison, both from monitoring and from modelling scenarios, about WPLreference and EbA policies.

RC2 - The authors should separate the results and discussion. Some sentences, for example lines 343-345;349-354;357-360; and so on, should be put into Discussion. The independent discussion could further clearly tell the readers your finding. **Answer:** We appreciate very much this comment. The adapted these new lines in order to help the reader abou our findings, but also not exceeding the limits of total words of the manuscript.

RC2 - in Section 3.6, the authors do not depict the results from Figure 17. **Answer:** We appreciate very much this review. In Section 3.6. We appended extra statements about the comparative results of Figure 17 in the new version of the paper as follows [because of the extension of these new statements, we suggest worth appending them in a Supplementary Material section, according to Editor final decision). "Figure 17 depicts a summary of monitored and modelled water yield obseervations and scenarios compared with EbA and GWF outputs in the catchments studied at the Cantareira System. The main bold, vertical, dotted line represents the regional mean water yield, compared with water yields from simulated scenarios, also including their respective GWFs. This figure clearly points six different conditions, labelled by letters (A, B, C, D, E and F), which configure potential scenarios of water security according to land-use change and insecurity thresholds, also showing tradoffs between water yield and grey water footprint outputs, explained in the text"

RC2 - delete the references from the conclusions. **Answer**: We corrected the conclusions, without citing any references.

RC2 - Table 1 should be moved to Supplemental information, or part of Table 1 should be merged in to Table 2. **Answer**: We appreciate this comment. We corrected and attended these suggestions, merging and realocating the tables.

RC2 - Table 8 should be moved in to Supplemental information. **Answer:** There is not a Table 8. Maybe Table 4(?). We have just moved it to Supplemental Material. Thank you.

RC2 - Fig.4, explain the meaning of the lines in the figure. **Answer**: Dotted lines represent trend lines for some selected basins here illustrated. Our interest in this figure was to question whether there would be both regional trend or scaling in the calibration coefficients, but not found in this first paper. Regional trends of the calibration can show both limits and uncertainty of modelling of complex catchments. Because of space, we decided to drop this figure out of the updated version.

**Fig.5,** RC2 - he sentence "Time (horizontal axis) is represented by month/year" is meaningless; further, to provide the meaning of the uncertaintybars andsample numbers. **Answer**: We appreciate this comment. We corrected the quotation. The uncertainty bars represent the minimum and maximum values of measured streamflow and pollutant loads in a cross section of the river during a field campaign of headwaters' catchments. The high variance in observations of field evidences explain the greater variability of these headwaters at the Cantareira System

**Fig.6,** RC2 – what are the meaning of the"size of circles" and the numbers? **Answer:** It is only a representation of a 3-D graph, substituting the 3rd axis with the diameter of the circle proportional to the magnitude of the 3rd variable (in this case, the 3rd variable is the turbidity). The number showed the value of turbidity. The figure 6 showed that, although a coherent and proportional relation existed in between observed mean river velocity and observed specific flow, experimental evidences still depicted outliers, from not only reference catchments with EbA/PES-Water options, but also intervention catchments with no EbA/PES-Water options, reflecting an illustrative example of how complex LULC options from EbA would be exaustively sensed into hydrological parameters and simulation scenarios. For those reasons, we adapted our conclusion and recommendations for further studies about new hypothesis' testing, according to fore-mentioned answers to reviewers.

[revised manuscript text omitted]

**Comentado [UdW12]:** Attending reviewer's comments, we moved this table to supplementary material of the paper, as Supplementary Table S.4

**Comentado [UdW13]:** Attending to comments of reviewer 1 and reviewer 2, table 5 was moved to Supplementary Material, as Supplement Table S.3

FIGURES

[revised manuscript text omitted]

Draft- Responses to Reviewers. Hydrol. Earth Syst. Sci. Discuss., https://doi.org/10.5194/hess-2017-474 Manuscript under review for journal Hydrol. Earth Syst. Sci  CC BY 4.0 License.

[Figure]

**Figure 16:** Hydrologic Service Index (circle ratio) related to drainage area of river basin (horizontal axis) and composite of water pollution index (vertical axis) for S2+EbA scenario: Equal weights of nitrate, total phosphorus and dissolved sediments are expressed in *WPLcomposite*.